# Integrated TORC1 and PKA signaling control the temporal activation of glucose-induced gene expression in yeast

Joseph Kunkel[1,2], Xiangxia Luo[1,2] & Andrew P. Capaldi[1]

The growth rate of a yeast cell is controlled by the target of rapamycin kinase complex I (TORC1) and cAMP-dependent protein kinase (PKA) pathways. To determine how TORC1 and PKA cooperate to regulate cell growth, we performed temporal analysis of gene expression in yeast switched from a non-fermentable substrate, to glucose, in the presence and absence of TORC1 and PKA inhibitors. Quantitative analysis of these data reveals that PKA drives the expression of key cell growth genes during transitions into, and out of, the rapid growth state in glucose, while TORC1 is important for the steady-state expression of the same genes. This circuit design may enable yeast to set an exact growth rate based on the abundance of internal metabolites such as amino acids, via TORC1, but also adapt rapidly to changes in external nutrients, such as glucose, via PKA.

---

[1] Department of Molecular and Cellular Biology, University of Arizona, Tucson, AZ 85721-0206, USA. [2]These authors contributed equally: Joseph Kunkel, Xiangxia Luo. Correspondence and requests for materials should be addressed to A.P.C. (email: capaldi@email.arizona.edu)

Cells respond to nutrient, stress, and hormone stimuli by activating complex signaling circuits. Determining how these circuits function is key to understanding cell behavior and should shed light on the way that cells integrate information and make decisions.

Experiments in the model organism, *Saccharomyces cerevisiae*, have led to the identification of a particularly interesting signaling circuit that includes the highly conserved target of rapamycin kinase complex I (TORC1) and cAMP-dependent protein kinase (PKA) pathways. In this circuit, the TORC1 and PKA pathways work together to regulate expression of the ribosome and protein synthesis genes[1–7] and, as a consequence, set the growth rate of the cell[1,2,8–10].

TORC1 induces expression of the ribosome and protein synthesis genes by phosphorylating and activating the transcriptional activator Sfp1[4,11,12] and the kinase Sch9[2,5,13] (Fig. 1a). Sch9 then phosphorylates and inactivates the transcriptional repressors Dot6/Tod6 and Stb3[2,5,14] (Fig. 1a). Meanwhile, the PKA kinases, Tpk1–3 (all close homologs of Sch9) work in parallel with TORC1 to activate Sfp1[4,12], and in parallel with Sch9 to phosphorylate and inactivate Dot6 and Tod6 (Fig. 1a)[2,5].

TORC1 also activates the PKA pathway via Sch9 dependent inhibition of Slt2—an event that limits the activity of the PKA inhibitor Bcy1[15,16]. However, this crosstalk appears to have little impact on gene expression[6,7,17].

While the TORC1 and PKA pathways activate a common set of genes, each pathway responds to different types of stimuli (Fig. 1a). Specifically, the PKA kinases are primarily activated by glucose and other fermentable sugars via Ras and Gpa2[1,3,18,19], while TORC1 is activated by nitrogen/amino acids and glucose/energy via the small GTPases Gtr1/2, the AMP activated protein kinase Snf1, and other unknown factors[1,20–25]. It has therefore been proposed that the TORC1 and PKA pathways work together to send distinct signals to the ribosome and protein synthesis genes[1,3,7]. However, this model does not explain how signals transmitted through the TORC1 and PKA pathways are integrated to set overall gene expression levels.

Here, to determine how the TORC1 and PKA pathways cooperate to regulate the ribosome and protein synthesis genes, we use DNA microarray analysis of yeast cells treated with chemical inhibitors, and carrying mutations, to follow mRNA levels and build a detailed model of the TORC1–PKA circuit that controls ribosome and protein synthesis gene expression. Our data show that the TORC1 and PKA pathways play distinct roles in the circuit, with the TORC1 pathway setting the steady-state level of gene expression and the PKA pathway primarily driving expression during transitions into, and out of, the rapid growth state in glucose. We propose that this circuit design helps the cell to set a growth rate based the level of internal metabolites such as amino acids via TORC1, but also adapt rapidly to changes in the concentration of the key external nutrient, glucose, via PKA.

## Results

**Pulsatile dynamics of gene expression.** To determine how the TORC1 and PKA pathways regulate cell growth, we set out to measure the impact that the TORC1 and PKA kinases have on gene expression in cells growing in glycerol (where PKA has low activity), and then at various time-points after cells are exposed to glucose (where PKA has high activity)[1,3,18,19].

As a first step, we measured the expression changes that occur when glucose (2%) is added to cells growing in glycerol. As expected[3,18], we found that glucose triggers massive changes in gene expression, including ≥3-fold upregulation of 574 genes, and ≥3-fold downregulation of 747 genes (Fig. 1b). Most of the genes upregulated in glucose are involved in ribosome and protein

synthesis (Ribi genes, Fig. 1b), while those downregulated in glucose tend to be involved in aerobic respiration (Fig. 1b).

Importantly, our expression data also show that the genes involved ribosome biogenesis and protein synthesis are highly expressed 20 min after glucose repletion (due to new transcription, Supplementary Fig. 1), but the expression levels drop once cells enter log phase growth at 120 and 180 min (Figs. 1b and 2a and Supplementary Fig. 2).

mRNA pulses—such as those found for the Ribi genes—have been shown to speed up protein synthesis by hyper-activating translation for a short period of time[26]. To explore this possibility, we followed the production of the Ribi gene *NSR1* during the glycerol to glucose transition using quantitative PCR and western blotting. In line with our microarray data (Supplementary Data file), *NSR1* mRNA was upregulated ~24-fold after 20 min in

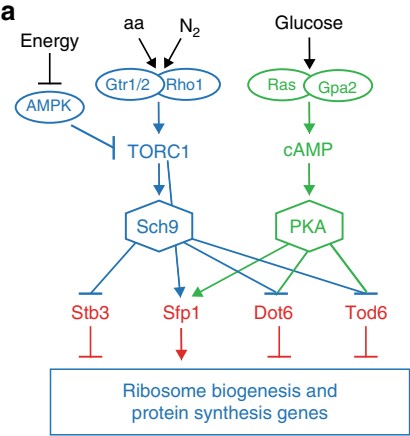

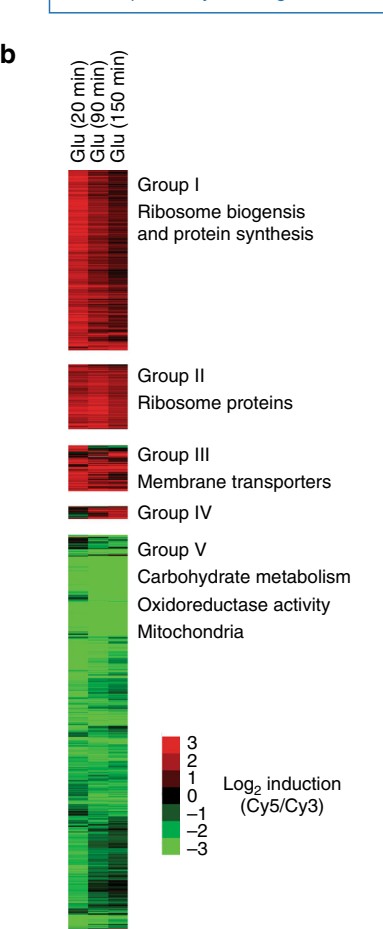

**Fig. 1** TORC1 and PKA signaling. **a** Working model of the TORC1–PKA signaling circuit. The TORC1 and PKA kinases regulate the transcriptional activator, Sfp1, and the transcriptional repressors, Stb3 and Dot6/Tod6, to control expression of the protein and ribosome synthesis genes, as described in the text. TORC1 and PKA also regulate numerous other factors involved in cell growth, including Maf1 (involved in Pol III regulation[48–50]) and Ifh1/Fhl1 (involved in ribosome protein gene expression[15,51,52]) not shown in this diagram. **b** Global response to glucose repletion. The gene expression program activated in glucose was measured by comparing the mRNA from Tpk1-3[as] cells growing in glycerol (labeled with Cy3) to the mRNA from Tpk1-3[as] cells exposed to 2% glucose for 20, 90, or 150 min (labeled with Cy5), on a two-color DNA microarray. Tpk1-3[as] cells have been shown to behave nearly identically to a wild-type strain in the absence of the Shokat inhibitor 1-NM-PP1[3], and were analyzed to allow comparison to other data in this study (Figs. 3 and 4). The heat map shows data for all genes with ≥3-fold activation or repression in glucose at one or more time-point. Hierarchical clustering led to the identification of five gene modules. Five hundred and seventy-four genes are induced in glucose, including (Group I) 341 genes involved in ribosome biogenesis and protein synthesis. This includes 116 genes involved in rRNA processing ($p < 1^{-100}$; Chi-square test with Benjamini correction) and 206 genes involved in ribosome biogenesis and assembly ($p < 1^{-100}$). (Group II) One hundred and twenty-three genes involved in ribosome function, including 100 components of the ribosome ($p < 1^{-100}$). (Group III) Eighty seven genes involved in a range of processes, including 21 transmembrane transporters ($p < 3^{-8}$), and 18 plasma membrane proteins ($p < 1^{-6}$). (Group IV) Twenty four genes involved in glycolysis ($p < 3^{-8}$) and alcohol biosynthesis ($p < 2^{-10}$). Seventy hundred and forty-seven genes are downregulated in glucose (Group V), including 93 genes involved in carbohydrate metabolism ($p < 8^{-40}$), 91 genes involved in oxidoreductase activity ($p < 9^{-30}$), and 215 genes involved in mitochondrial function ($p < 1^{-19}$). The gene ontology groups were identified using GO Stat[53]

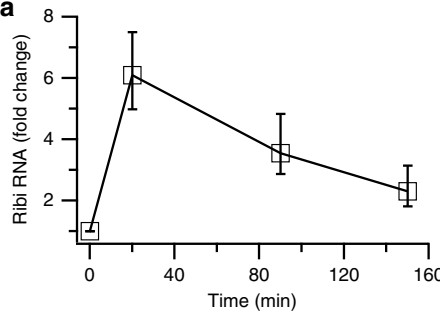

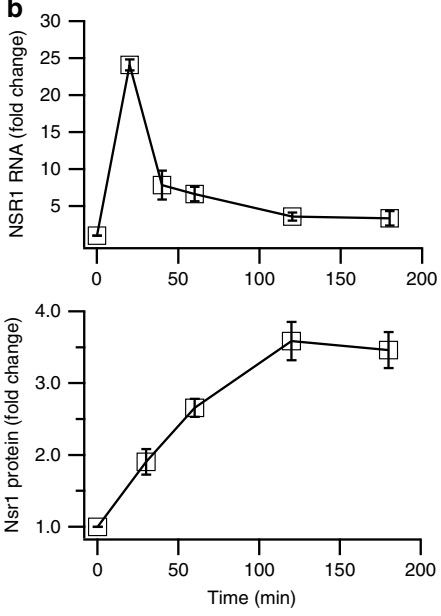

glucose, but was only 3.6 ± 0.6 and 3.3 ± 1.0-fold (errors are the SD for three biological replicates) above baseline levels after 120 and 180 min in glucose (Fig. 2b). In contrast, Nsr1 protein levels increased monotonically to a steady-state level around 3.5-fold above the baseline (Fig. 2b). We also saw similar behavior when we followed the level of two other Ribi proteins, Dhr2, and Mrd1 (Supplementary Fig. 4)

To see how the *NSR1* mRNA and protein expression data fit together, we built a simple mathematical model. The mRNA dynamics were simulated using a short (square-wave) pulse of RNA production followed by a smaller, sustained increase in the transcription rate, with a single rate constant for mRNA decay. Protein levels were then modeled using standard first order kinetics, with a single rate constant for translation ($k_3$) and a single rate constant for protein decay ($k_4$).

To constrain our model, we measured the rate of Nsr1 decay in cells treated with cycloheximide, using a western blot. In line with previous reports showing that Ribi proteins are among the most stable in yeast[27], we found that Nsr1 levels remained steady over a 4-h period (Supplementary Fig. 3c). We therefore set the rate constant for Nsr1 decay so that it matched the doubling rate of our strain in SD medium ($k_4 = 0.08$ min$^{-1}$), to account for Nsr1 loss due to dilution. Finally, we set the only free variable in the model—the rate constant for translation ($k_3$)—so that the final concentration of Nsr1 protein matched that found in our experiments. This simple model shows that a large pulse in *NSR1* mRNA should, indeed, translate into a monotonic increase in Nsr1 protein levels with an apparent rate constant that roughly matches our observations (red solid and dotted lines, Fig. 2c).

Altering the parameters in our model showed that reducing the intensity of the mRNA pulse leads to a dramatic increase in the

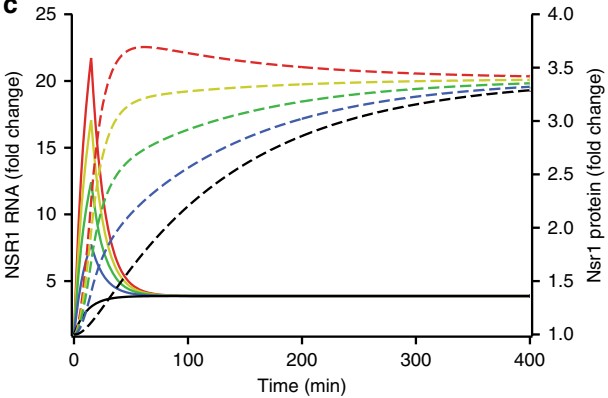

**Fig. 2** Dynamics of protein and ribosome gene expression. **a** Average expression of Ribi genes from Group I in Fig. 1 (for top 300 out of 341 genes). Error bars show the expression of genes at the 25th and 75th percentiles. **b** Fold change in *NSR1* mRNA expression (top panel) and Nsr1 protein level (bottom panel) after introduction of glucose to cells growing in glycerol, as measured by qPCR and western blotting, respectively (see Supplementary Fig. 3 for raw data). The data show the average and standard deviation from three biological replicates, normalized based on the Nsr1 protein or RNA levels at time zero. **c** Simulations of *NSR1* mRNA (solid lines) and protein (dotted lines) synthesis as described in the text for different levels of mRNA production (color coded)

time it takes Nsr1 protein to reach its final steady-state level (yellow, green, blue, and black lines; Fig. 2c). In contrast, the final steady-state level of Nsr1 protein only depends on the final steady-state level of the *NSR1* mRNA.

Thus, our data and modeling indicate that the glucose dependent induction of the ribosome biogenesis genes includes two stages: (1) a large transient pulse of mRNA expression that serves to drive rapid protein production, and (2) a smaller sustained increase in mRNA expression that sets, and maintains, the final protein levels.

**Expression component analysis**. To measure the impact that the TORC1 and PKA kinases have on the two phases of the glucose response, we employed expression component analysis[28,29]: In the first experiment, yeast carrying mutations that render all three PKA kinases sensitive to the Shokat Inhibitor 1-NM-PP1 (Tpk1-3[as])[3] were grown to log phase in glycerol and treated with 1-NM-PP1 to block PKA activity, rapamycin to block TORC1 activity, rapamycin + 1-NM-PP1 to block TORC1 and PKA activity, or drug carrier alone (DMSO). The cultures were then harvested after 30 min and the mRNA from cells treated with each drug, or drug combination, compared with the mRNA from cells treated with DMSO on a two-color DNA microarray. In the second experiment Tpk1-3[as] cells were grown to log phase in glycerol, treated with rapamycin, 1-NM-PP1, rapamycin and 1-NM-PP1, or DMSO for 10 min (to ensure the TORC1 and/or PKA pathway were inactive), and 2% glucose added to each culture. The cells were then harvested after 20 min, and the gene expression changes measured, again comparing the mRNA from each drug-treated culture to the mRNA from the DMSO control. In the final two experiments, Tpk1-3[as] cells were grown to log phase in glycerol, 2% glucose added to the medium, and the cells treated with drugs or DMSO after 60 or 120 min. The cells were then harvested 30 min later and the mRNA levels in the drug and DMSO-treated samples compared.

The microarray data were then used to split the expression data for each gene, at each time-point, into three components; TOR, the activation from TORC1 in the absence of PKA; PKA, the activation from Tpk1-3 in the absence of TORC1; and Co, the activation that results from an interaction (cooperation) between TORC1 and PKA. This was done by comparing the $\log_2$ expression change found in rapamycin, 1-NM-PP1, and rapamycin + 1-NM-PP1, to Eqs. (1–3) describing the components lost in each drug treatment/microarray experiment (bottom of Fig. 3a). Specifically:

(1) Rapamycin treatment leads to a loss of both TORC1 signaling and any interaction between TORC1 and PKA (TOR + Co components).

(2) 1-NM-PP1 treatment leads to a loss of both PKA signaling and any interaction between TORC1 and PKA (PKA + Co components).

(3) Rapamycin + 1-NM-PP1 treatment leads to a loss of all PKA and TORC1 signaling (TOR + PKA + Co components).

Thus, we calculated the TOR component by subtracting the expression defect in 1-NM-PP1 (PKA + Co) from the expression defect in 1-NM-PP1 + rapamycin (TOR + PKA + Co); the PKA component by subtracting the expression defect in rapamycin (TOR + Co) from the expression defect in 1-NM-PP1 + rapamycin (TOR + PKA + Co); and the Co component by subtracting the value of the TOR and PKA components (calculated as above) from the expression defect in 1-NM-PP1 + rapamycin (TOR + PKA + Co). Note that in this system of equations the expression components and microarray data have the opposite sign (since TOR activity goes down in rapamycin, etc).

Our expression component analysis provides a comprehensive (global) view of TORC1 and PKA-dependent transcriptional control during glucose repletion, including regulation of

metabolism and nutrient transporters (Supplementary Fig. 5). However, we focused our analysis on the ribosome biogenesis, protein synthesis and ribosomal protein gene modules (Groups I and II, Fig. 1b), since their expression is tightly coupled to cell growth[1,8–10], and depends on a well-understood portion of the TORC1–PKA circuit described in the introduction.

Remarkably, we found that 461/463 ribosome and protein synthesis genes are regulated ≥2-fold by both TORC1 and PKA if you include data from all four time-points in our study (Fig. 3b and Supplementary Data file). However, the influence that the TORC1 and PKA kinases have on gene expression changes dramatically during the glycerol to glucose transition.

First, in glycerol medium, TORC1 acts on its own to induce gene expression (TOR component only, glycerol columns, Fig. 3b). This makes sense, since the PKA pathway has very low activity in the absence of glucose or other fermentable sugars[1,3,18,19].

Next, 20 min after cells are exposed to glucose, most genes are activated by both TORC1 and PKA, but there is very little cooperation between the two factors (305/463 genes activated by TOR and PKA, at a twofold cutoff, Fig. 3b and Supplementary Data file). This expression pattern (PKA and TOR components, but no Co component) shows that the TORC1 and PKA kinases act independently (in parallel) during initial response to glucose and that the PKA kinases drive the transient pulse of gene expression (middle panel, Fig. 3a).

Finally, as cells reach their steady-state growth rate, at 90 and 150 min (Supplementary Fig. 2), the influence that PKA has on gene expression changes again. At a subset of ribosome biogenesis genes, PKA-dependent induction disappears almost entirely, so that gene expression is regulated by TORC1 alone (Group I b, Fig. 3b). At other genes (Groups I a and c, Fig. 3b), TORC1 and PKA stop acting independently, and start acting cooperatively (Co component only; top panel, Fig. 3a) or partially cooperatively (TOR + Co component; lower panel, Fig. 3a). The PKA kinases also induce expression of the ribosome protein genes at 90 and 150 min (Group II, Fig. 3b), but PKA only activates these genes in the absence of TORC1 activity (i.e., PKA activity is redundant with TORC1 activity, thus PKA + Co = 0).

**TORC1 and PKA cooperate to regulate Dot6/Tod6**. The expression component data show that the TORC1 and PKA pathways cooperate to regulate gene expression during log growth in glucose (150 min time-point, Fig. 3b). To gain insight into how this occurs, we compiled a list of genes activated/repressed by each transcription factor in the ribosome and protein synthesis control circuit (Dot6/Tod6, Stb3, and Sfp1) based on published data[4,12,30,31], and compared the list to our TORC1–PKA expression component data (Fig. 4a).

Our analysis revealed that Dot6 and Tod6 act at most genes induced cooperatively by TORC1 and PKA (Co component, Fig. 4a), but few genes regulated by TORC1 alone (Fig. 4a). We also found a DNA sequence recognized by Dot6/Tod6 (known as a PAC site[32]) in the promoter of many genes with a cooperative expression component (Fig. 4a). In contrast, the transcriptional activator Sfp1 and the transcriptional repressor Stb3 (known to bind to the RRPE motif[14,33]), act at genes regulated by TORC1 alone and genes regulated cooperatively by TORC1 and PKA (Fig. 4a). Based on these correlations, and previous data showing that TORC1 and PKA phosphorylate and inactivate Dot6/Tod6 in log growth conditions[2,5], we hypothesized that Dot6/Tod6 are required for cooperation between the TORC1 and PKA pathways.

To test this idea, we performed expression component analysis in *dot6Δtod6Δ* cells. The resulting data show that deletion of Dot6/Tod6 completely eliminates cooperation

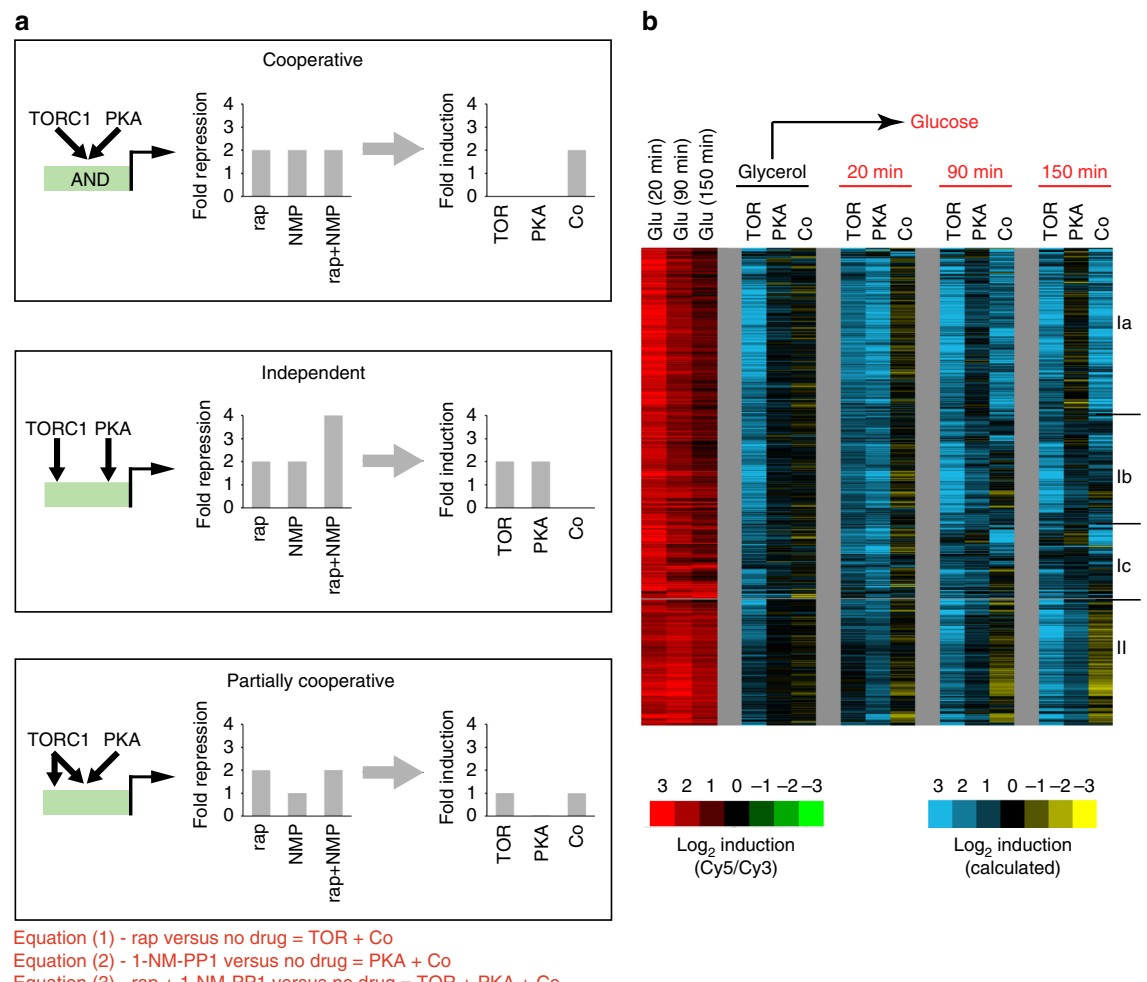

Equation (1) - rap versus no drug = TOR + Co
Equation (2) - 1-NM-PP1 versus no drug = PKA + Co
Equation (3) - rap + 1-NM-PP1 versus no drug = TOR + PKA + Co

**Fig. 3** Expression component analysis. **a** Expression component analysis was used to determine if, and how, TORC1 and PKA cooperate to regulate gene expression as cells transition from growth in glycerol to growth in glucose. To see how expression component analysis works it is useful to compare three possible scenarios. In the first scenario (upper panel), the TORC1 and PKA kinases are both absolutely required for activation of a particular gene. In this case, the expression defect in rapamycin (due to TORC1 inhibition) is equal to the expression defect in 1-NM-PP1 (due to PKA inhibition) and the expression defect in rapamycin + 1-NM-PP1 (due to TORC1 and PKA inhibition). Solving Eqs. (1–3), then leads us to a positive Co component and no TOR or PKA component. In the second scenario (middle panel), the TORC1 and PKA kinases act independently (in parallel) to activate the same gene, leading to a positive TORC1 and PKA component, but no Co component (Co = 0). Finally, in the third scenario (lower panel), the TORC1 and PKA kinases act cooperatively to activate the same gene (for example, if both factors phosphorylate and activate a single transcription factor) but TORC1 also activates the gene through a separate mechanism, leading to positive TORC1 and Co components, but no PKA component (PKA = 0). Expression component analysis can also be used to identify other modes of regulation, including redundancy (OR gating) where the Co component has a negative value[28,29]. **b** Expression component analysis of TORC1 and PKA signaling in glycerol and glucose medium. The first three columns in the heat map show the raw expression data for the ribosome and protein synthesis genes (Groups I and II), taken from Fig. 1. The remaining columns show the expression components calculated using the rapamycin, 1-NM-PP1, and rapamycin + 1-NM-PP1 data listed in Supplementary Data file and the equations listed in part (a), as explained in the text. Groups I a, b, and c were defined by clustering the expression component data for Group I from Fig. 1. All experiments shown here have been repeated at least twice, leading to near identical expression component data

between the TORC1 and PKA pathways, as well as most PKA signaling to the ribosome and protein synthesis genes (12/463 genes with ≥3-fold activation by PKA + Co, Fig. 4b and Supplementary Data file). In contrast, TORC1 continues to activate gene expression on its own in cells missing Dot6 and Tod6 (220/463 genes with ≥3-fold activation by TOR, Fig. 4b and Supplementary Data file). Therefore, we conclude that the TORC1 and PKA pathways work together to inactivate the transcriptional repressors Dot6 and Tod6 during log phase growth (Co component), and that TORC1 also acts through a Dot6/Tod6 independent mechanism to regulate many of the same genes (TOR component).

**TORC1 and PKA can act independently of Dot6/Tod6.** Dot6/Tod6 play a critical role in TORC1 and PKA signaling during log phase growth, but how do Dot6/Tod6 impact the initial stages of the glucose response? To answer this question, we examined the expression components in *dot6Δtod6Δ* cells exposed to glucose for 20 min (Fig. 4c). Surprisingly, we found that deletion of Dot6/Tod6 does not impact gene expression in the presence of TORC1 and/or PKA inhibitors (Fig. 4c). We also found that the initial response to glucose repletion occurs independently of Sfp1 and Stb3 (Supplementary Fig. 6), indicating that the PKA and TORC1 pathways act through a distinct set of (currently unknown) factors when cells are first exposed to glucose.

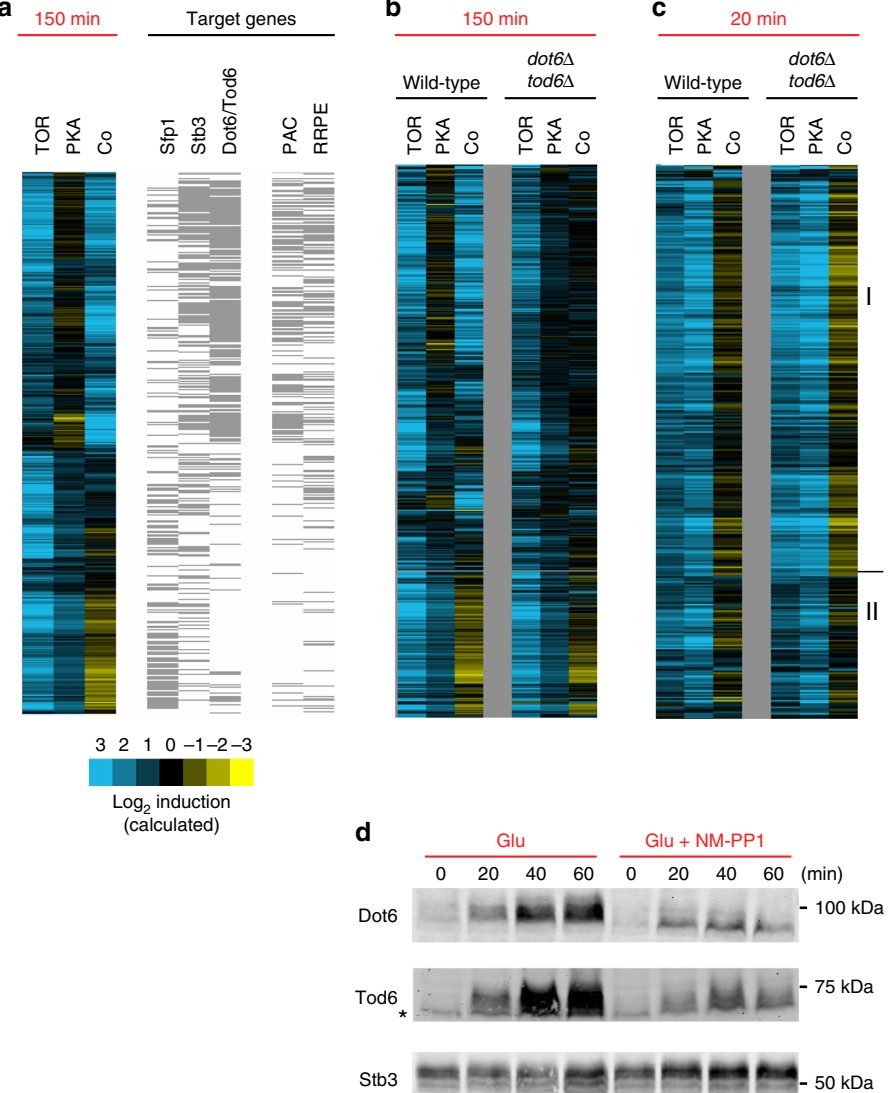

**Fig. 4** The role of Dot6/Tod6 in TORC1–PKA signaling. **a** Comparison of the TORC1–PKA expression components from Fig. 3 (left three columns) to a list of genes activated/repressed by each transcription factor in the TORC1–PKA circuit. The left set of gray bars shows the genes activated by Sfp1 (>2-fold change in the microarray data of reference[12]), repressed by Stb3 (>2-fold change in the microarray data of reference[31]), and repressed by Dot6/Tod6 (>2-fold in the microarray data of 30). The right set of gray bars shows the genes that have Dot6/Tod6-binding site (PAC site) or Stb3-biding site (RRPE site) in their promoter (as listed in 4). **b** and **c** Expression component analysis in *dot6Δtod6Δ* cells. Data are shown next to the expression component data from Fig. 2 for comparison. **d** Western blot analysis to examine Dot6, Tod6, and Stb3 levels during the glycerol to glucose transition. Tpk1-3[as] cells with HA-tagged Dot6, Tod6, or Stb3, were grown in glycerol medium as described in the "Methods" section, and cells harvested before, or at various time-points after, 2% glucose ±1-NM-PP1 was added to the culture. All experiments shown here were repeated at least three times with near identical results. The asterisk symbol indicates the band seen in glycerol lane of the Tod6 western blot is also present in blots examining extracts from wild-type cells and is therefore not due to low levels of Tod6-HA in glycerol medium

**Rewiring of the TORC1–PKA circuit in glucose**. Our expression component data show that Dot6 and Tod6 have little to no influence on gene expression during the initial response to glucose (even when PKA and/or TORC1 are inhibited), but then play an important role in gene regulation during log growth in glucose. But how does this switch from Dot6/Tod6 independent regulation, to Dot6/Tod6 dependent regulation, occur? We reasoned that Dot6/Tod6 levels might be low in glycerol and then increase in glucose.

To test this hypothesis, we first looked to see if Dot6/Tod6 transcripts are produced at a higher level in glucose than in glycerol. This was not the case: Dot6 and Tod6 mRNA levels are similar during steady-state growth in glycerol and glucose (<2-fold difference, Supplementary Data file). We then asked if the

Dot6/Tod6 protein levels change in glucose. To do this we performed western blot analysis to examine the levels of HA-tagged versions of both Dot6 and Tod6 during growth in glycerol, and at several time-points after cells are exposed to glucose (using Stb3 as a control). Strikingly, we found that both Dot6 and Tod6 are very low during growth in glycerol, and accumulate to near maximum levels 40 min after the cells are exposed to glucose (Fig. 4d). Furthermore, full accumulation of Dot6 and Tod6 in glucose requires the activity of the PKA pathway itself (Fig. 4d), indicating that the PKA pathway not only plays a key role in regulating the ribosome and protein synthesis genes, it also drives the rewiring of the TORC1–PKA circuit by triggering accumulation of the repressor proteins Dot6 and Tod6 in glucose.

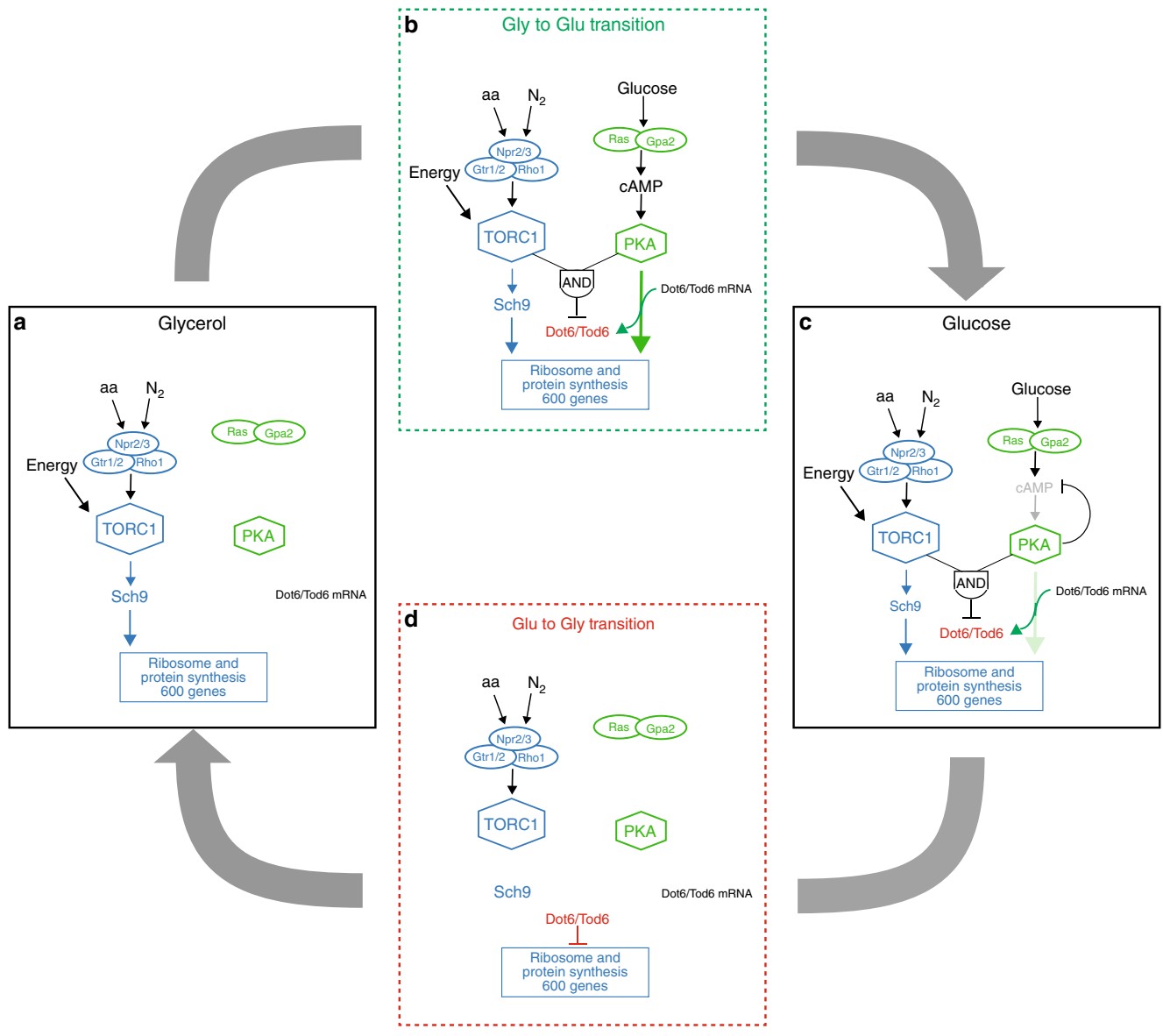

**Fig. 5** Model of the TORC1–PKA signaling circuit. **a** The TORC1 pathway promotes expression of the protein and ribosome synthesis genes (and thus cell growth) in glycerol medium. **b** Immediately after cells are exposed to glucose, the PKA kinases act in parallel with the TORC1 pathway to drive a pulse of ribosome and protein synthesis gene expression. **c** Then, as cells reach steady-state growth in glucose, the gene expression levels are reduced (due to negative feedback in the PKA pathway) so that they are about 2.5-fold higher than in glycerol. **d** Once glucose is depleted, the repressor proteins Dot6/Tod6 (synthesized due to activation of the PKA pathway, see **b**) are dephosphorylated due to inactivation of TORC1 and PKA, move to the nucleus, and recruit Rpd3L to the protein and ribosome synthesis genes. This causes dramatic downregulation of the ribosome and protein synthesis genes until Dot6/Tod6 are degraded (30–60 min later), and the circuit returns to its original (glycerol growth) state. See text for further details

**The TORC1-PKA-dependent transcriptional circuit**. Putting our data together, we are able to construct a detailed model of the TORC1–PKA circuit. This model describes TORC1 and PKA signaling as cells transition from slow growth in glycerol, to rapid growth in glucose, and back again (Fig. 5):

(A)  In glycerol medium, the TORC1 pathway acts largely on its own to drive expression of the protein and ribosome synthesis genes (Fig. 3b). The PKA pathway has low activity in synthetic medium with glycerol due to the absence of glucose and other fermentable sugars known to activate Ras and Gpa2 (Fig. 1a)[1].

(B)  Twenty minutes after cells are exposed to glucose, the ribosome and protein synthesis genes are upregulated by an average of 5.7-fold (Fig. 2a and Supplementary Data file).

This dramatic increase in gene expression is driven by the PKA kinases, acting in parallel with TORC1 (Fig. 3b), and does not occur when cells are transferred from medium containing glucose and a poor nitrogen source (proline), to medium containing glucose and the optimum nitrogen source (glutamine)—conditions known to activate TORC1 (Supplementary Fig. 7). In fact, PKA kinase activity accounts for over 75% of gene induction observed during the initial pulse phase of glycerol to glucose transition (Supplementary Data file). These data fit well with previous studies from the Broach and Heideman labs showing that the PKA pathway is necessary[3,6] and sufficient[18] for most of the early transcriptional response in glucose repletion.

Importantly, our results also show that exposure to glucose triggers PKA-dependent upregulation of Dot6 and Tod6 protein levels (Fig. 4d).

(C)    As cells reach steady-state growth in glucose at 90/150 min, the expression of most ribosome and protein synthesis genes decreases again, so that the average expression level is just 2.5-fold above that found in glycerol (Fig. 2a and Supplementary Data file). This decrease in gene expression is due, at least in part, to a decrease in PKA pathway activity. First, there is a dramatic decrease in PKA-dependent gene induction at almost 100 genes as cells transition to log phase growth (Fig. 3b). Second, at many other ribosome and protein synthesis genes we see a switch, the PKA pathway stops acting independently and starts working with TORC1 to regulate genes through Dot6/Tod6 (Fig. 3b). This switch is especially important because Dot6 and Tod6 are inactive repressors during log phase growth in glucose[2,5]. In other words, the PKA kinases do not have a major impact the absolute level of gene expression during log phase growth; they just act with TORC1 to keep Dot6 and Tod6 phosphorylated and inactive. In contrast, TORC1 continues to induce gene expression through a Dot6/Tod6 independent mechanism during log phase growth.

While our data provides strong evidence that PKA pathway activity decreases as cells adapt to glucose, our experiments do not help to explain how this happens. However, previous studies show that concentration of the PKA kinase activator, cAMP, increases fivefold a few minutes after yeast cells are exposed to glucose and then decreases again over the next 5–10 min until it reaches a steady-state concentration twofold above the prestimulation level[34]. This decrease in cAMP concentration is caused by negative feedback; the PKA kinases activate the phosphodiesterases Pde1 and Pde2 to clear cAMP from the cell, and likely inactivate the upstream factor Ras to limit cAMP synthesis[34,35]. It therefore seems likely that PKA kinase activity decreases as cells adapt to glucose, due to negative feedback, and this stops the PKA kinases from activating the ribosome and protein synthesis genes. However, residual activity in the PKA pathway is sufficient to promote synthesis of Dot6 and Tod6, and is needed (along with TORC1 activity) to keep Dot6 and Tod6 phosphorylated and inactive.

(D)    The final step in the TORC1-PKA-dependent response to glucose stimuli is the return to the glucose starvation, or glycerol growth, state. The Broach and Loewith labs have already examined this transition in detail to show that glucose starvation (and other stress/starvation stimuli) trigger activation of Dot6 and Tod6, and dramatic downregulation of the ribosome and protein synthesis genes via the HDAC Rpd3L[2,5]. What our study adds to this picture is the discovery that it is glucose and the PKA pathway that trigger the accumulation of Dot6/Tod6. As a consequence, once cells are starved for glucose Dot6 and Tod6 are degraded (~30 min half-life, Supplementary Fig. 8) so that the system returns to a state where TORC1 acts on its own to regulate gene expression.

Putting all the steps in the glycerol to glucose growth transition together, we see that the PKA pathway helps drive the cell into a rapid growth state when cells are first exposed to glucose (Fig. 5b) by triggering hyper-activation of the ribosome and protein synthesis genes and thus rapid synthesis of the encoded proteins. The PKA pathway also helps ensure that the cell is driven to a slow growth state when cells are starved for glucose, via Dot6 and Tod6 (Fig. 5d). In contrast, the TORC1 pathway plays the major role in setting the absolute level of Ribi gene (and protein) expression during steady-state growth in glycerol and glucose (Fig. 5a, c).

## Discussion

Previous studies of TORC1 and PKA signaling have revealed that the TORC1 and PKA kinases regulate many of the same transcription factors and genes[1–6,12]. However, it was unknown how the TORC1 and PKA pathways work together to control gene expression levels. Here, using chemical inhibitors and expression component analysis, we were able to take snapshots of the TORC1–PKA signaling system at different time-points and build a detailed model of the TORC1–PKA signaling circuit. Our model shows that the PKA and TORC1 pathways play distinct roles in gene regulation; the PKA pathway primarily drives expression of the ribosome and protein synthesis genes during the transition into, and out of, the rapid growth state in glucose, while the TORC1 pathway plays the major role in setting gene expression levels during steady-state growth.

But why does the TORC1–PKA circuit function in this way?

Previous work suggests that the TORC1 pathway acts (at least in part) as a feedback controller: TORC1 drives the use of energy and amino acids by activating protein and ribosome synthesis, translation and other processes[1,5]. However, if energy or amino acid levels fall, TORC1 signaling is repressed via AMPK or the Rag proteins Gtr1/2[20,21,23,25,36,37].

This kind of feedback control is known to have two advantages[38]. First, it is very accurate; the rate of protein synthesis will always be set based on the level of resources available to the cell. Second, it works in the face of any perturbation, predictable or not.

Importantly, however, feedback control modules also have a limitation, they respond poorly (slowly) to large external changes[39–41]. As a result, man-made control systems often include a feedforward control module that drives the response to key input signals, and then lets the more accurate feedback controller take over again[39–41]. This is precisely how the PKA pathway functions. When glucose levels increase, PKA drives the TORC1 dependent gene expression program to a new, higher level. Then, (presumably) via negative feedback, this PKA signaling channel turns mostly off again. Similarly, when the cells are growing rapidly in glucose, PKA stabilizes Dot6/Tod6 so that a loss of glucose triggers rapid and dramatic downregulation of the ribosome and protein synthesis genes. However, over time, Dot6/Tod6 are degraded and the TORC1 pathway regulates gene expression and mass accumulation primarily on its own.

This hybrid feedback/feedforward control model explains the role that the TORC1 and PKA pathways play in regulating gene expression during transitions into and out of glucose, but does not fully explain the response to nitrogen stimuli. Specifically, we find that transferring cells from a poor nitrogen source (proline) to a high-quality nitrogen source (glutamine) triggers a small pulse of Ribi gene expression that is completely dependent on TORC1 activity, and only partially dependent on PKA activity (Supplementary Fig. 7). Thus, it appears that the TORC1 pathway is transiently hyper-activated when cells are first exposed to a high-quality nitrogen source, and consequently that TORC1 has a role outside of setting steady-state gene expression levels (at least in response to a nitrogen upshift). Further work is therefore needed to test the hybrid feedback/feedforward control model in a range of conditions and explore its implications.

It will also be important to map transcription factor binding to the ribosome and protein synthesis genes over time to determine exactly how the TORC1 and PKA pathways control gene expression. For example, we expect to see an increase in Dot6/

Tod6 binding at the Ribi genes during the initial response to glucose starvation and then a loss of Dot6/Tod6 binding over time as the factors are degraded. However, it is less clear how Sfp1 and Stb3 binding will change over time, or how these factors will influence gene expression, especially in light of a recent study showing that Sfp1 can bind promoters both directly and through a cofactor[42].

While numerous questions remain about the intricacies of the TORC1/PKA gene regulatory circuit, the data presented here for the PKA pathway and protein/ribosome synthesis genes are clear: the PKA kinases mainly drive gene expression during transitions into and out of glucose, but have little influence steady-state gene expression levels. Going forward, it will be interesting to see if other nutrient response and/or hormone signaling pathways act in a similar way.

## Methods

**Strains**. The base strain used in this study, Tpk1-3[as], was created using a loop in/loop out procedure[3]. First, *TPK1*, *TPK2*, and *TPK3* were cloned into a pRS306 vector. Next, a Quickchange reaction (Agilent) was used to introduce a Shokat mutation into each plasmid (to create Tpk1[M164G], Tpk2[M147G], and Tpk3[M165G]. The three plasmids were then cut using BseRI, BlpI, and AflII, respectively, and transformed into both a and alpha type cells; W303 strains EYO690 and 691 from the O'Shea lab (trp1 leu2 ura3 his3 can1 GAL+ psi+). Ura+ cells were then struck on 5FOA media to select for mutants that had looped out the Ura3 marker, and therefore carry a single copy of Tpk1, 2, or 3. Finally, these strains were mated to create the Tpk1-3[as] strain. At each step in this procedure we sequenced in and around the Tpk1, 2, and 3 genes to ensure the strain selected carry the Shokat mutation(s), and no other change.

Once we had the Tpk1-3[as] strain, the *dot6Δtod6Δ* strain was produced by transforming the cells with PCR products that consist of auxotrophic markers, flanked by the 40 bp sequence found directly up and downstream from the gene, and selection on the appropriate medium[43].

HA-tagged (Dot6-HA, Tod6-HA, Stb3-HA, Mrd1-HA, and Dhr2-HA) strains were constructed similarly but the primers directed the recombination to the sites directly upstream and directly downstream from the stop codon[44]. PCR was used to confirm the location of the marker gene or epitope tag insertion. Sequencing was used to check the integrity of the HA tagged and deletion strains.

For experiments measuring the transcriptional response to glutamine (Supplementary Fig. 7), the Tpk1-3[as] strain was modified by knocking in the wild-type, functional forms of His3, Leu2, Trp1, and Ura3 at their native loci so that the cells could growth with proline (or glutamine as the only nitrogen source).

All strains used in this study are available upon request and a list of the primers used in strain construction is found in Supplementary Table 1.

**DNA microarrays**. In the glucose induction time-course experiment, an overnight culture of TPK1-3[as] cells was used to inoculate three 4 L conical flasks containing 1 L of SC + 3% glycerol media to an OD$_{600}$ of 0.05. These cultures were then grown, shaking at 200 rpm and 30 °C, to an OD$_{600}$ of 0.3 before 500 mL of cells were collected by vacuum filtration and frozen in liquid nitrogen. At this point glucose was added to the remaining flasks (to a 2% final concentration), and 150 OD units of cells was collected (as above) at each time point (20, 90, 150 min). mRNA was then extracted from the cells using hot phenol, purified using a poly A sepharose column, and converted to aa-UTP-labeled complementary DNA (cDNA) using StrataScript II reverse transcriptase (NEB). The cDNA was then labeled with Cy3 or Cy5, and transcript levels measured using Agilent (G4813A) DNA microarrays, an Axon 4000B scanner, and GenPix 6.1 software[28,29].

In the expression component analysis experiments, an overnight culture of TPK1-3[as], or TPK1-3[as] *dot6Δtod6Δ* cells, was used to inoculate two 4 L conical flasks containing 1 L of SC + 3% glycerol medium to an OD$_{600}$. These cultures were then grown, shaking at 200 rpm and 30 °C, to an OD$_{600}$ of 0.4 and then 375 mL of cells were collected, as described above. The remaining cells were transferred to create four 400 mL cultures, each in a 2.8 L conical flask. Glucose was added to each flask (to 2% final concentration) and the cultures grown for 20, 60, or 120 min. Cultures were treated for 30 min with DMSO, 1-NM-PP1 (100 nM final concentration), rapamycin (200 ng/L final concentration), or both 1-NM-PP1 and rapamycin, before 150 OD units of cells collected by vacuum filtration and frozen in liquid nitrogen. Cells collected at 20 min were treated with inhibitors for 10 min before the addition of glucose, to insure the PKA and TORC1 pathways were locked off before the glucose response was activated. Expression analysis was then carried out as described above, for the time-course experiments, except that drug treated cells were labeled with Cy5 while cells without treatment were labeled with Cy3. Cells were exposed to 1-NM-PP1 and/or rapamycin for 30 min to ensure that there was enough time for the mRNA to degrade and expose the impact of losing the TORC1 and/or PKA-dependent activation. Specifically, most RNAs have a half-life of >10[45], so we should pick up at least and eightfold change at genes completely

regulated by TORC1/PKA. However, this time is short enough to limit the impact that failing to activate/repress key genes has on gene expression profile (i.e., secondary effects)[29].

**Western blots**. To measure the level of Dot6, Tod6, and Stb3 proteins, an overnight culture of a Dot6-HA, Tod6-HA, or Stb3-HA cells was used to inoculate two 1 L flasks containing 250 mL of SC + 3% glycerol to an OD$_{600}$ of 0.1. These cultures were grown, shaking at 200 rpm and 30 °C, to an OD$_{600}$ of 0.6 and a 47 mL sample was collected, mixed with 3 mL 100% trichloroacetic acid (TCA), and held on ice for at least 30 min. One flask was treated with 1-NM-PP1 (100 nM final concentration) for 10 min, and then glucose was added to each flask to a 2% final concentration. 47 mL samples were then collected and treated with TCA, as before, at 20, 40, and 60 min time-points. TCA treated samples were centrifuged at 4000 rpm for 5 min at 4 °C, and the cell pellets washed twice with 4 °C water, twice with acetone, and disrupted by sonication at 15% amplitude for 5 s before centrifugation at 6000 g for 30 s. Cell pellets were then dried in a speedvac 10 min at room temperature, and frozen until required at −20 °C. Protein extraction was performed by bead beating (6 × 1 min, full speed) in urea buffer (6 M Urea, 50 mM Tris-HCl pH 7.5, 5 mM EDTA, 1 mM PMSF, 5 mM NaF, 5 mM NaN$_3$, 5 mM NaH$_2$PO$_4$, 5 mM *p*-nitrophenylphosphate, 5 mM β-glycerophosphate, and 1% SDS) supplemented with complete protease and phosphatase inhibitor tablets (Roche 04693159001 and 04906845001). The lysate was collected after centrifugation for 5 min at 2000 g, resuspended into a homogenous slurry by vortexing, and heated at 65 °C for 10 min. Soluble proteins were then separated from insoluble cell debris by centrifugation at 21,000 g for 5 min, and the lysates stored at −80 °C until required. Cell extracts were then heated to 95 °C in SDS sample buffer for 5 min before they were run on an SDS-PAGE gel, transferred to a nitrocellulose membrane, and the tagged protein detected using a 12CA5 (anti-HA; Roche 11 867 423 001) primary at 1:1000 and goat anti-mouse secondary labeled with IRDy 800CW (LiCor 925–32219) at 1:5000, and imaged a LiCor Odyssey.

To measure the level of Nsr1, wild-type cells were grown, fixed with TCA, and processed as described above, except that the glucose was added to the culture at an OD$_{600}$ of 0.1 so that the cells remained in log phase for the full 4-h experiment and 15 OD units of cells were harvested at each time-point. Nsr1 was detected using anti-Nsr1 from Thermo (MA1-10030) at a 1:2000 dilution, and goat anti mouse conjugated to IRDye 800CW at a 1:5000 dilution. We confirmed that this antibody recognizes Nsr1 using an *nsr1Δ* strain (Fig. S3b). Dhr2-HA and Mrd1-HA were detected using 12CA5 as described above. Blots were also probed with anti-Tap42 from Santa Cruz (sc-67113) at 1:500 and goat anti-rabbit labeled with IRDye680 (LiCor 926–32221) at 1:10,000. The values reported are all based on the ratio of the Nsr1, Dhr2, or Mrd1 (green) signal to the Tap42 (red) loading control. Examples of uncropped and unprocessed images are provided in Supplementary Fig. 9.

**NSR1, DHR2, and MRD1 qPCR**. NSR1, DHR2, or MRD1 mRNA levels were measured using TaqMan probes recognizing NSR1, DHR2 or MRD1, and Pex6 (a house keeping gene). The yeast were grown and harvested as described for the microarray experiments, except that the cells were washed into *RNAlater* stabilization solution before freezing. The mRNA was then extracted from the cells using hot phenol, treated with DNase I, and converted to cDNA using StrataScript II reverse transcriptase. qPCR reactions were then run in Perfecta pPCR ToughMix (low Rox, Quanta Bio) using Taqman probes and primers (Lifetech, probe and primer sequences not provided by company) on an Agilent MX3005p cycler[46]. The expression values reported are all based on the ratio of the NSR1, DHR2, or MRD1 to PEX6 signal in each PCR reaction.

To measure the level of nascent transcripts (Supplementary Fig. 1), we immunoprecipitated Rpb3-3xFLAG and measured NSR1, DHR2, or MRD1 mRNA levels (each along with a PEX6 control) using taqman primers and probes that bind inside the first 400 bp of the transcript[47].

**Modeling**. Nsr1 mRNA and protein dynamics were modeled in Berkeley Madonna using the following equations (1000 min simulation).

The input signal driving gene expression was modeled as a square pulse with two parts

$$S = X * squarepluse(0, 15) + Y * squarepluse(0, 1000) \quad (4)$$

where $X = 1.0$ and $Y = 0.15$

RNA and Protein ($P$) dynamics were then modeled using simple first order differential equations:

$$d/dt(\text{RNA}) = S * k_1 - \text{RNA} * k_2 \quad (5)$$

$$d/dt(P) = k_3 * \text{RNA} - k_4 * P \quad (6)$$

with initial RNA and protein concentrations of 1

$k_1 = 2.2$, $k_2 = 0.085$, $k_3 = 0.007$, and $k_4 = 0.008$ all in min$^{-1}$

To explore the impact that limiting the size of the transient burst in mRNA production has on protein synthesis (Fig. 1e) the value of $X$ in Eq. (4) was reduced from 1.0 (red line) to 0.75 (yellow line), 0.5 (green line), 0.25 (blue line) and 0.0 (black line), while all other values were held constant.

**Reporting Summary**. Further information on research design is available in the Nature Research Reporting Summary linked to this article.

## Data availability

All data generated and analyzed during this study are included in this article and its supplementary information files. The microarray data are also available in the GEO database, accession number GSE133591.

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

## Acknowledgements

We acknowledge financial support from NIH T32 GM084905 (JK), NSF DGE-0654435 (JK), and NIH 1R01GM097329 (APC). We would also like to thank, Ted Weinert, Tricia

Serio, Rod Capaldi, and members of the Capaldi lab for critical reading of the manuscript.

## Author Contributions

A.P.C. and J.K. designed the experiments, analyzed the data, and wrote the paper. J.K. and X.L. built the strains and performed the experiments.

## Additional information

**Competing interests:** The authors declare no competing interests.

