## [Peer Review File · Nature Communications]

Reviewers' comments:

Reviewer #1 (Remarks to the Author):

I reviewed this manuscript several years ago and the following is the first half of my review, which still applies:

"The manuscript from Kunkel et al addresses the interplay of different signaling pathways regulating transcriptional changes in yeast, particularly those of the ribosomal biogenesis (RIBI) genes, following the transition from growth on a poor carbon source to growth in glucose. Using now standard techniques of microarray-based measurements of global transcriptional changes in strains in which the protein kinase A (PKA) and TORC1 pathways can be inactivated by exogenous application of inhibitors, the investigators determined the extent to which ribosomal biogenesis gene induction and maintenance depend on signaling through PKA, TORC1 or both. These studies have yielded a fascinating and credible model in which TORC1 is responsible for maintenance of RIBI gene expression, whereas PKA is required for transition from glycerol to glucose and, in cooperation with TORC1, for maintenance of expression during growth on glucose. Moreover, the investigators identify Dot1/Tod1 as the locus for collaborative activity of PKA and TORC1 and propose a new locus for TORC1 regulation of basal RIBI gene expression. The investigators integrate all of these data into a sophisticated model of RIBI gene regulation in transitions between rich and poor growth conditions. These experiments are well executed and interpreted and the results quite compelling. Moreover, these studies will be of interest not only to investigators studying yeast growth control but also to those interested in means of dissecting integrated signaling networks as well as those interested in the role of TORC1 in growth control."

At the time I objected to the poor description of the methodological and computational approaches used by the investigators. Some of that concern still remains but the presentation is much better. In addition, the authors add an intriguing aspect of Dot1/Tod1 regulation at the post-transcriptional level that responds to glucose. This adds an additional and intriguing layer to the overall mechanism of the homeostatic regulation of RIBI gene expression and growth control.

However, I have a serious concern regarding the experimental approach adopted by the authors and the effect that might have on their model of regulation. In particular, their experiment rests on adding glucose to cultures growing on a poor carbon source but replete with ammonium, a rich nitrogen source. Accordingly, under the initial conditions, the PKA pathway is essentially off while the TORC1 pathway is quite active. They then shift to a rich carbon source with the same nitrogen source, which activates the PKA pathway but doesn't alter TORC1 signaling. Perhaps not surprisingly, they find that PKA contributes primarily to the initial alteration in transcription while TORC1 has a longer term effect. The question then becomes what would be the outcome if they started with a culture grown in glucose on a poor nitrogen source and then shifted to a rich nitrogen source. Would their analysis suggest that TORC1 was responsible for rapid restructuring while PKA contributed more to the homeostatic regulation? Such a result would indicate that their separation of function of the two pathways was not a fundamental difference in the contribution of the two pathways to short term versus long term control but simply reflected what initial conditions were interrogated. Without that experiment, the authors' model is interesting but not comprehensive.

Minor points:

p. 3, end of paragraph 3: "inactivate" is more accurate than "repress."

p. 4, log phase growth begins at 50 min, according to Fig S1. Expression is measured to 120 and 180 but that doesn't tell us when expression levels dropped.

Figure 2A, the graph should be labeled with "1NM-PP1" rather than "N".

Reviewer #2 (Remarks to the Author):

This manuscript describes experiments in which steady-state mRNA measurements are used in the context of chemical inhibition of two key growth regulatory pathways in yeast (TORC1 and PKA) to deconstruct that mechanisms and kinetics underlying the cellular response to glucose. Although this work largely confirms conclusions regarding TORC1 and PKA action made by the Loewith and Broach & Heideman labs, respectively, the “expression component analysis” employed here to deconvolute the independent and combined roles of the two pathways has led to a more detailed picture and one new insight. Specifically, the authors present evidence that the combined action of TORC1 and PKA operates at least in part through the regulation of Dot6/Tod6, two repressors whose protein levels are up-regulated by glucose in a PKA-dependent manner to influence ribosome protein and Ribi gene expression.

Critique

Although in general the experiments described here are well designed, a major problem is the absence of a more direct measure of transcription, for example by 4-thio uracil (4-tU) labeling, NET-seq, or RNA polymerase II ChIP-seq. Because they only measure steady-state mRNA levels the authors cannot distinguish between changes in transcription initiation/elongation and changes in mRNA stability following glucose addition. As recent studies have demonstrated a remarkable buffering effect (i.e. opposing changes in initiation and mRNA stability following various genetic perturbations) it would be important to determine to which extent the changes measured here are actually due to changes in initiation, as the authors appear to assume. This point is fundamental to many of the conclusions in this work.

A second deficiency is the absence of any direct evidence for an increase in Dot6/Tod6 action at ribosomal protein and Ribi target genes during the transition for high to steady-state transcription levels as cells adapt to the glucose pulse. Since this proposed feed-back role of Dot6/Tod6 represents the key novel finding here, it warrants further study, for example by ChIP-seq monitoring of Dot6/Tod6 or Rpd3/Sin3 binding. In addition, the authors should directly test the role of Pde1/2 in their proposed feed-back model. One possibility would be to rapidly deplete these proteins, using the auxin-induced degron system, during the glucose response.

Additional points

1. Introduction pg. 3: « TORC1.....phosphorylating and activating the transcriptional activator Sfp1 ». Which, if any, of the referenced papers actually shows that TORC1 phosphorylates Sfp1 directly?
2. Introduction pg. 3: “we use DNA microarray analysis”. The authors should state explicitly what they measure.
3. Results pg. 4: There is no reference for PKA being inactive in glycerol (same in Results pg.6)
4. Results, Fig 1. It is not clear why Tpk1-3as has been used as “wild type” strain as it is not. This should be explained, particularly since at this point in the text the reader does not know what this strain is, since it is not introduced until the following paragraph.
5. Results, Fig. 1D: it might be better to integrate the Western Blot into the figure. It would also be clearer if the time scales were made identical for C-E, perhaps using “//” to indicate the break between 200 and 400 min in E.
6. Results: the author should extend the analysis made for NSR1 in Fig. 2 to other RIBi genes (at least another one). They already have the microarray data; it would be enough just to do the protein analysis by Western Blot. Furthermore, is the effect on translation specific for Ribi proteins or are other proteins also affected?
7. Details related to the calculated induction analysis (Fig. 2B; 3A-C) should be more clearly explained

in the text. For example, it is unclear how the cluster analysis was performed and what the authors believe are the distinguishing features of the different groups, some of which appear to be quite heterogeneous (particularly groups III and V).

8. Similarly, it is unclear what distinguishes groups Ia, b, c (Fig. 2B) and why this distinction is not carried forward in Fig. 3A-C.

9. Results pg.5: the description of the experiment is not clear. A small panel with a scheme showing when glucose and inhibitors are added would be helpful. The description in the figure legend is also confusing.

10. Results pg. 7, beginning pg.8: The section on Dot6 and Tod6 is somewhat confusing and not so convincing. Is it really surprising that their deletion has no effect on the glucose response considering that they work as transcriptional repressors?

11. Results pg. 7: references for the published data used should be included in the main text.

Reviewer #3 (Remarks to the Author):

In this study, Kunkel et al. investigate the role of the TOR and PKA signaling pathways in inducing gene expression changes in yeast switched from glycerol to glucose. The authors show that upon this switch, there is a rapid "pulse"-like increase in mRNA encoding growth genes (RiBi/protein synthesis genes), followed by a rapid decrease to a steady-state mRNA level that maintains a higher steady-state protein level. Using expression components analysis, the authors determine that the initial increase in mRNA is mediated by TOR and PKA signaling acting independently, while in the steady-state condition, these pathways act cooperatively or partially cooperatively, without a substantial independent PKA component. Finally, the authors demonstrate that Dot6/Tod6 is required to mediate cooperation between TOR and PKA signaling in this latter steady-state condition. The authors propose that such a regulatory mechanism would permit rapid adaptation to changes in growth conditions, while longer-term steady-state protein synthesis rates are set by TOR signaling based on nutrient conditions.

This study is very interesting, in that it provides substantial insights into how the relative contribution of, and interactions between, these two key signaling pathways evolve over time following growth substrate switching. However, there are several issues that should be further considered.

1. In undertaking expression components analyses, the authors split gene expression changes into three components: TOR, PKA, and Co (i.e. changes that require both TOR "AND" PKA signaling) (Fig. 2). However, there could be a fourth component that arises from TOR "OR" PKA signaling, in other words, expression changes that only require one of TOR or PKA signaling, but not necessarily both. In practice, this component could be of importance and contribute substantially if TOR and PKA regulate the same downstream factors (see Fig. 1a), and in substantially identical ways such that signaling from either arm is sufficient to saturate signaling through this downstream factor. This could impact the components calculated, e.g. in treating with rapamycin, TOR + Co would be lost, but PKA + OR component would remain, not just the PKA component as noted. The authors should consider assessing this OR component, particularly as it is not a truly "independent" component of each of the two pathways, but, in the current model, is included within the TOR and PKA "independent" components.

2. The authors derive expression components based on expression changes following drug treatment at various time points following switch to glucose (Fig. 2). The authors should consider including the corresponding expression heatmaps from which these calculations are based. Fig. S3 provides a similar insight. However, it is notable that at 20 min, the independent TOR (and PKA) components contribute substantially (Fig. 2b) whereas "rap vs no drug" elicits relatively little expression change compared to the other treatments (Fig. S3). This is contrary to expectation based on the current model.

3. Both TOR and PKA act on gene expression through phosphorylation of known downstream substrates (Fig. 1a). The authors infer based on drug treatment effects that these pathways contribute to different extents over time. The authors should consider examining changes in downstream phosphorylation markers under these same treatments to assess whether changes in these markers correlate with the expression components inferred by their mathematical model.

MINOR COMMENTS:

4. The authors use NSR1 as a general example of RiBi genes. It may be worthwhile to examine a few other RiBi genes (e.g. RPS or RPL proteins), to confirm on a per gene level that the trend on protein and RNA is replicated, and to examine degree of variation observed. It is worth noting that NSR1 RNA changes might be much higher (approx. 25-fold at peak) than the average RiBi mRNA (approx. 6-fold at peak) (Fig. 1C, 1D), although part of this may be due to the limited dynamic range of microarray measurements.

5. The authors assess RNA levels after 20 min or 30 min drug treatment (Fig. 2, S3), and use this to infer relative contribution of respective pathways. Could the authors provide justification that this is an appropriate length of drug treatment, e.g. perhaps based on RNA degradation rates or similar considerations?

Reviewers' comments:

Reviewer #1 (Remarks to the Author):

I reviewed this manuscript several years ago and the following is the first half of my review, which still applies:

“The manuscript from Kunkel et al addresses the interplay of different signaling pathways regulating transcriptional changes in yeast, particularly those of the ribosomal biogenesis (RIBI) genes, following the transition from growth on a poor carbon source to growth in glucose. Using now standard techniques of microarray-based measurements of global transcriptional changes in strains in which the protein kinase A (PKA) and TORC1 pathways can be inactivated by exogenous application of inhibitors, the investigators determined the extent to which ribosomal biogenesis gene induction and maintenance depend on signaling through PKA, TORC1 or both. These studies have yielded a fascinating and credible model in which TORC1 is responsible for maintenance of RIBI gene expression, whereas PKA is required for transition from glycerol to glucose and, in cooperation with TORC1, for maintenance of expression during growth on glucose. Moreover, the investigators identify Dot1/Tod1 as the locus for collaborative activity of PKA and TORC1 and propose a new locus for TORC1 regulation of basal RIBI gene expression. The investigators integrate all of these data into a sophisticated model of RIBI gene regulation in transitions between rich and poor growth conditions. These experiments are well executed and interpreted and the results quite compelling. Moreover, these studies will be of interest not only to investigators studying yeast growth control but also to those interested in means of dissecting integrated signaling networks as well as those interested in the role of TORC1 in growth control.”

At the time I objected to the poor description of the methodological and computational approaches used by the investigators. Some of that concern still remains but the presentation is much better. In addition, the authors add an intriguing aspect of Dot1/Tod1 regulation at the post-transcriptional level that responds to glucose. This adds an additional and intriguing layer to the overall mechanism of the homeostatic regulation of RIBI gene expression and growth control.

We thank the reviewer for his/her kind comments.

However, I have a serious concern regarding the experimental approach adopted by the authors and the effect that might have on their model of regulation. In particular, their experiment rests on adding glucose to cultures growing on a poor carbon source but replete with ammonium, a rich nitrogen source. Accordingly, under the initial conditions, the PKA pathway is essentially off while the TORC1 pathway is quite active. They then shift to a rich carbon source with the same nitrogen source, which activates the PKA pathway but doesn't alter TORC1 signaling. Perhaps not surprisingly, they find that PKA contributes primarily to the initial alteration in transcription while TORC1 has a longer term effect. The question then becomes what would be the outcome if they started with a culture grown in glucose on a poor nitrogen source and then shifted to a rich nitrogen source. Would their analysis suggest that TORC1 was responsible for rapid restructuring while PKA contributed more to the homeostatic regulation? Such a result would indicate that their separation of function of the two pathways was not a fundamental difference in the contribution of the two pathways to short term versus long term control but simply reflected what initial conditions were interrogated. Without that experiment, the authors' model is interesting but not comprehensive.

This is a good point. To address this issue, we replaced all of the auxotrophic markers in the Tpk1-3as strain (His3, Leu2, Trp1 and Ura3) with the wild-type (functional) genes at their native loci. We then measured the expression of three Ribi genes (NSR1, MRD1 and DHR2) in the new strain during the transition from log growth in synthetic medium + glucose with proline as the only nitrogen source, to log growth in synthetic medium + glucose with glutamine as the only nitrogen source (Fig. S6). This shift—from a poor nitrogen source to the optimum nitrogen source—led to a 1.7-fold increase in the growth rate. Despite this, there is only a small/short pulse of Ribi gene expression as the cells adapt to the new growth medium. Moreover, a substantial fraction of the pulse is PKA dependent—in line with work showing that the PKA pathway is activated at a low level when amino acids are added back to nitrogen starved cells (1). Thus, we can conclude that the PKA pathway is the major driver of the pulse in gene expression observed when cells are transferred from poor to rich nutrient sources, and that this pulse is far more dramatic when cells are transferred into glucose containing medium (the known major activator of the PKA pathway). This data fits well with the model we present in the paper.

Minor points:

p. 3, end of paragraph 3: “inactivate” is more accurate than “repress.”

We agree and have changed the wording

p. 4, log phase growth begins at 50 min, according to Fig S1. Expression is measured to 120 and 180 but that doesn't tell us when expression levels dropped.

This is a fair point but with the microarrays we were focused on capturing what goes on during the pulse of gene expression and then once cells reach steady state growth. However, we now include qPCR data, following the mRNA levels of DHR2, MRD1 (as well as the NSR1 data) with good time resolution, and from these data it is clear that the pulse of gene expression lasts until around 120 min.

Figure 2A, the graph should be labeled with “1NM-PP1” rather than “N”.

Yes, N could be taken to mean nitrogen (particularly with the inclusion of the new data). We have changed it to NMP (rather than NM-PP1) so that the labels do not take up too much space.

Reviewer #2 (Remarks to the Author):

This manuscript describes experiments in which steady-state mRNA measurements are used in the context of chemical inhibition of two key growth regulatory pathways in yeast (TORC1 and PKA) to deconstruct that mechanisms and kinetics underlying the cellular response to glucose. Although this work largely confirms conclusions regarding TORC1 and PKA action made by the Loewith and Broach & Heideman labs, respectively, the “expression component analysis” employed here to deconvolute the independent and combined roles of the two pathways has led to a more detailed picture and one new insight. Specifically, the authors present evidence that the combined action of TORC1 and PKA operates at least in part through the regulation of Dot6/Tod6, two repressors whose protein levels are up-regulated by glucose in a PKA-dependent manner to influence ribosome protein and Ribi gene expression.

Critique

Although in general the experiments described here are well designed, a major problem is the absence of a more direct measure of transcription, for example by 4-thio uracil (4-tU) labeling, NET-seq, or RNA polymerase II ChIP-seq. Because they only measure steady-state mRNA levels the authors cannot distinguish between changes in transcription initiation/elongation and changes in mRNA stability following glucose addition. As recent studies have demonstrated a remarkable buffering effect (i.e. opposing changes in initiation and mRNA stability following various genetic perturbations) it would be important to determine to which extent the changes measured here are actually due to changes in initiation, as the authors appear to assume. This point is fundamental to many of the conclusions in this work.

This is a good point. While it seems logical that the cell would limit RNA production in poor nutrient conditions, and increase RNA production when nutrients are abundant (rather than produce excess RNA and degrade it in starvation conditions), it is important to demonstrate that is the case. Therefore, we introduced a 3XFLAG tag at the C-terminus of Rbp3 in our background strain, and following the protocol of Churchman and Weissman, immunoprecipitated RNA Pol II and measured the level of the associated (nascent) RNA transcripts using qPCR rather than sequencing since our goal was simply to confirm that there is a large increase in newly synthesized RNA after the cells are transferred from glycerol to glucose medium (rather than mapping RNA Pol stall sites etc). The data show that there is a large increase in nascent RNA transcripts at all three genes we examined; 15-fold at MRD1 and 40-fold at NSR1 and DHR2 (three genes we have now use as test cases throughout the paper). Thus, the gene induction seen in the glycerol to glucose transition is at least mostly due to new transcription.

A second deficiency is the absence of any direct evidence for an increase in Dot6/Tod6 action at ribosomal protein and Ribi target genes during the transition for high to steady-state transcription levels as cells adapt to the glucose pulse. Since this proposed feed-back role of Dot6/Tod6 represents the key novel finding here, it warrants further study, for example by ChIP-seq monitoring of Dot6/Tod6 or Rpd3/Sin3 binding.

There appears to be some confusion about our model, the role of feedback, and the role that Dot6 and Tod6 play in TORC1/PKA circuit. Dot6 and Tod6 do not drive the decrease of Ribi gene expression as cells adapt to glucose. Instead, our data show the following: When cells are first exposed to glucose, the PKA pathway is hyperactive and drives high level expression of the Ribi genes (Figs. 1 and 2). Then as cells reach their steady state growth rate, PKA activity decreases, so that PKA only acts to keep Dot6 and Tod6 *inactive* (Fig. 2). That is Dot6 and Tod6 are phosphorylated and inactive during growth in glucose (2, 3) and thus do not drive the decrease in Ribi gene expression. This is clear from the data in Fig. 3 where we show that the impact that PKA has on gene expression decreases dramatically at 90 and 150 min compared to that at 20 min, even in a strain missing Dot6/Tod6 (Fig. 3). Instead inhibition of the PKA pathway over time (via known feedback mechanisms) dampens gene activation that occurs via yet unknown transcription factors (pg 7 and Fig. S7).

So, when do Dot6 and Tod6 matter? As explained at the end of the paper, our data show that Dot6 and Tod6 accumulation in glucose serves to modify the circuit so that when TORC1 or PKA inhibition does occur (in starvation conditions, or in TORC1/PKA inhibitors), the Dot6/Tod6 repressor proteins are dephosphorylated and trigger strong down regulation of Ribi gene expression. This drives rapid repression of the Ribi genes to speed up the transition from rapid growth to slow growth. Then, over time, Dot6/Tod6 are lost so that the Ribi gene expression becomes entirely independent of PKA again, and TORC1 sets the growth rate.

Given our model and data, it is not useful for us to measure Dot6/Tod6 and/or Rpd3 binding. It is very well established that Dot6/Tod6 bind and repress the Ribi genes when you starve cells growing in glucose (2-4). What we are adding here is the finding that Dot6/Tod6 are absent, and do not influence gene expression (even when Dot6/Tod6 are inactivated by inhibitors), when the cell grows in the absence of glucose for an extended period of time. A ChIP experiment would just confirm that transcription factors that are absent from the cell, and have no impact on gene expression, are not binding their target genes.

We apologize for any confusion about the role of Dot6/Tod6 in the TORC1/PKA circuit. We have now added several statements on page 7 and 8 (in the sections on component analysis) to make it clear that the impact of PKA is to keep the repressor proteins Dot6/Tod6 inactive, and that the role of Dot6/Tod6 is only apparent when the cells are treated with TORC1 and PKA inhibitors (three highlighted sentences on pgs 7-8).

In addition, the authors should directly test the role of Pde1/2 in their proposed feed-back model. One possibility would be to rapidly deplete these proteins, using the auxin-induced degron system, during the glucose response.

We have shown that PKA dependent activation of Ribi gene expression decreases dramatically as cells adapt to glucose so that TORC1 ultimately takes over and sets the steady state rate of Ribi and RP gene expression. It is well known that there are multiple negative feedback loops (involving both Pde1/2 and Ira1/2) in the PKA pathway—and this explains why PKA activity (and cAMP levels) pulse when cells are exposed to glucose. Breaking the feedback loops without altering pre and post stimulation levels of cAMP—levels that also depend on Pde1/2 and Ira1/2—would be very difficult, if not impossible, especially given the short timescale of the cAMP pulse (a few minutes). Moreover, it is out of the scope of the work described in this paper. We have built a detailed model of the way that TORC1 and PKA cooperate to control gene expression in response to nutrients (with major implications for the way complex and parallel signaling circuits function), and have shown that the role of PKA in gene induction is lost as cells reach their steady state growth rate, but are not trying to determine exactly how PKA activity levels are set at this stage.

Additional points

1. Introduction pg. 3: « TORC1.....phosphorylating and activating the transcriptional activator Sfp1 ». Which, if any, of the referenced papers actually shows that TORC1 phosphorylates Sfp1 directly?

Lempianinen et al (reference 11) shows that TORC1 directly phosphorylates Sfp1.

2. Introduction pg. 3: “we use DNA microarray analysis”. The authors should state explicitly what they measure.

This is now changed to “we use DNA microarray analysis of yeast cells treated with chemical inhibitors, and carrying mutations, to follow mRNA levels and build a detailed model of the TORC1-PKA circuit that controls ribosome and protein synthesis gene expression”

3. Results pg. 4: There is no reference for PKA being inactive in glycerol (same in Results pg.6)

We have added those reference numbers and also changed the words to say low activity and high activity. We were careful to say low activity and high activity elsewhere in the paper since cAMP and PKA activity levels have not been measured accurately enough to say they are on or off (just low vs high).

4. Results, Fig 1. It is not clear why Tpk1-3as has been used as “wild type” strain as it is not. This should be explained, particularly since at this point in the text the reader does not know what this strain is, since it is not introduced until the following paragraph.

We used Tpk1-3as as the “wild-type strain” so that we can compare the data across experiments. Broach and co-workers (5) have already shown that the analog sensitive strain behaves very similarly to a standard wild-type strain, and we have not seen any significant difference between the data we acquired in this paper and those where we have studied TORC1 and PKA dependent gene expression in a standard wild-type previously (4, 6, 7). We have now added text to the figure legend to make the rationale behind our use of the Tpk1-3as strain clear.

5. Results, Fig. 1D: it might be better to integrate the Western Blot into the figure. It would also be clearer if the time scales were made identical for C-E, perhaps using “//” to indicate the break between 200 and 400 min in E.

We appreciate the comment, it would be ideal if you could directly compare the timescales. But the simulation looks very messy when you break the scale and the figure looks busy (detracting from the key points) when you include the western blot. It is also important to mention that the mathematical model is simplified (with a square wave of mRNA production and a single rate constant for mRNA decay and a single rate constant for protein decay) since its purpose is to unravel the influence that a pulse of mRNA production has on the rate of protein production when the protein is long lived, and not to exactly capture the dynamics of protein production. We have put the western blots directly below the new (supplemental) figures examining production of Dhr2 and Mrd1 to make them easier to view (Fig. S4).

6. Results: the author should extend the analysis made for NSR1 in Fig. 2 to other RiBi genes (at least another one). They already have the microarray data; it would be enough just to do the protein analysis by Western Blot. Furthermore, is the effect on translation specific for Rib proteins or are other proteins also affected?

We have now extended the analysis in Fig. 1d to two additional ribosome biogenesis genes and proteins; DHR2/Dhr2 and MRD1/Mrd1 (now Figure S4) and the data shows that they behave in a similar way to Nsr1. The paper is focused on picking apart the circuit that controls ribosome biogenesis gene and protein production, so we did not examine the synthesis rate of other proteins.

7. Details related to the calculated induction analysis (Fig. 2B; 3A-C) should be more clearly explained in the text. For example, it is unclear how the cluster analysis was performed and what the authors believe are the distinguishing features of the different groups, some of which appear to be quite heterogeneous (particularly groups III and V).

The the expression groups are distinguished by their expression behavior in Fig. 1, and were identified by hierarchical clustering (groups were just split off based on the dendrogram). Group I contains genes that have strong induction at 20min that falls away quickly at 90 and 150min. Group II contains genes with the strong induction at 90min, that falls away at 150 min, group IV has a delay turning on and group III includes genes that are induced but do not fit in with the other groups. Genes in group V are repressed in glucose. We could have broken the genes in Groups III and V down further but they are not the subject of this paper and thus we simply present the component data in the supplement. We have now modified the figure legend to say that hierarchical clustering was used.

A detailed explanation of the component analysis in Fig. 2 and 3 has been published previously (8, 9). This paper gives an approximately 2-page summary of expression component analysis (pg 5 and 6, Fig. 2a and Fig. 2a legend), we feel that anything longer and more in depth would be redundant with the previous papers and detract from the TORC1/PKA focused message in the paper.

8. Similarly, it is unclear what distinguishes groups Ia, b, c (Fig. 2B) and why this distinction is not carried forward in Fig. 3A-C.

Groups I and II are carried through from Fig. 1 to Fig. 2 as described in the Fig. 2 legend and below:

Group I was broken into subgroups by clustering the expression component data. Group 1a has TOR and Co components at steady state (90 and 150min), 1b has TOR but little to no Co component, while 1c has little TOR

component. These groups are not critical to the conclusions we draw but help the reader see that at some genes the PKA component switches to the Co component as the cells adapt to glucose (groups 1a and 1c) while at other genes the influence of PKA disappears altogether (group 1b)-- as mentioned on the bottom of page 6 and top of 7. We have now added several lines to the end of the Fig. 2b legend to make the differences between groups 1a, b and c clear.

The distinction was not carried through in Fig. 3 because it was much easier to see the match between the different transcription factor binding sites and the expression component behavior, when all of the data was clustered together. That is the point of Fig. 3a is to show that the genes with a cooperative component tend to be regulated by Dot6 and Tod6.

9. Results pg.5: the description of the experiment is not clear. A small panel with a scheme showing when glucose and inhibitors are added would be helpful. The description in the figure legend is also confusing.

This is a good point. We have now rewritten the section at the bottom of page five to carefully go through the timing of adding glucose and inhibitors, and emphasize the way that two-color DNA microarrays were used to measure the exact impact that TORC1 and PKA have on gene expression.

10. Results pg. 7, beginning pg.8: The section on Dot6 and Tod6 a somewhat confusing and not so convincing. Is it really surprising that their deletion has no effect on the glucose response considering that they work as transcriptional repressors?

It is important to note that the expression component analysis is based on measuring the impact that inhibiting TORC1 with rapamycin and/or inhibiting PKA with NM-PP1 has on gene expression. When TORC1 or PKA are inhibited during steady state growth (150 min), Dot6 and Tod6 are dephosphorylated triggering strong repression of the Ribi genes (this is results in the large TOR and Co components). In contrast, in a strain missing Dot6 and Tod6, inhibition of TORC1/ PKA has a much smaller impact on gene expression (Fig. 3b). When this same experiment is repeated in cells only exposed to glucose for 20 min (Fig. 3c) deletion of Dot6 and Tod6 does not influence gene expression (because Dot6 and Tod6 are not present in the cell). This discovery is fundamental to our new circuit model, since the loss of Dot6/Tod6 as cells starve for glucose allows TORC1 to control steady state gene expression in slow growing cells.

As mentioned earlier, we are sorry for any confusion about the role that Dot6/Tod6 play as repressors that are inhibited by TORC1 and PKA, and have modified and inserted three sentences in the section on Dot6/Tod6 function on pages 7 and 8 (highlighted in yellow) to make sure it is clear that (1) TORC1 and PKA phosphorylate the Dot6/Tod6 repressor proteins so that they are inactive when TORC1 and PKA are active and (2) we are measuring the impact of Dot6/Tod6 in the component analysis because we are measuring gene expression changes in TORC1 and/or PKA inhibitors (e.g. when the Dot6/Tod6 repressor proteins are active).

11. Results pg. 7: references for the published data used should be included in the main text.

We have added the references in the figure legend into the main text

Reviewer #3 (Remarks to the Author):

In this study, Kunkel et al. investigate the role of the TOR and PKA signaling pathways in inducing gene expression changes in yeast switched from glycerol to glucose. The authors show that upon this switch, there is a rapid "pulse"-like increase in mRNA encoding growth genes (RiBi/protein synthesis genes), followed by a rapid decrease to a steady-state mRNA level that maintains a higher steady-state protein level. Using expression components analysis, the authors determine that the initial increase in mRNA is mediated by TOR and PKA signaling acting independently, while in the steady-state condition, these pathways act cooperatively or partially cooperatively, without a substantial independent PKA component. Finally, the authors demonstrate that Dot6/Tod6 is required to mediate cooperation between TOR and PKA signaling in this latter steady-state condition. The authors propose that such a regulatory mechanism would permit rapid adaptation to changes in growth conditions, while longer-term steady-state protein synthesis rates are set by TOR signaling based on nutrient conditions.

This study is very interesting, in that it provides substantial insights into how the relative contribution of, and interactions between, these two key signaling pathways evolve over time following growth substrate switching.

We thank the reviewer for his/her kind comments.

However, there are several issues that should be further considered.

1. In undertaking expression components analyses, the authors split gene expression changes into three components: TOR, PKA, and Co (i.e. changes that require both TOR "AND" PKA signaling) (Fig. 2). However, there could be a fourth component that arises from TOR "OR" PKA signaling, in other words, expression changes that only require one of TOR or PKA signaling, but not necessarily both. In practice, this component could be of importance and contribute substantially if TOR and PKA regulate the same downstream factors (see Fig. 1a), and in substantially identical ways such that signaling from either arm is sufficient to saturate signaling through this downstream factor. This could impact the components calculated, e.g. in treating with rapamycin, TOR + Co would be lost, but PKA + OR component would remain, not just the PKA component as noted. The authors should consider assessing this OR component, particularly as it is not a truly "independent" component of each of the two pathways, but, in the current model, is included within the TOR and PKA "independent" components.

Expression component analysis does identify and quantify OR gating, but OR gating leads to a negative Co component, not a fourth component. To see this, it is best to work through the example of a gene that has perfectly redundant regulation by TORC1 and PKA (a pure, digital, OR gate). In this example, treatment with 1-NM-PP1 does not impact on gene expression because TORC1 activates the gene in the absence of PKA. Similarly, treatment with rapamycin does no impact gene expression because PKA activates the gene in the absence of TORC1. However, when the cell is treated with both rapamycin and 1-NM-PP1 the gene would turn off—in this case lets say expression decreases 16 fold or \log_2 of -4.

Following the equations Fig. 2a,

- (1) Change in rapamycin = TOR + Co = 0
- (2) Change in 1-NM-PP1 = PKA + Co = 0
- (3) Change in rap + NM-PP1 = TOR + PKA + Co = -4

Solving for PKA by subtracting equation 1 from 3 we get PKA = 4

Solving for TOR by subtracting equation 2 from 3 we get TOR = 4

Finally, plugging the values for TOR or PKA into equations 1 or 2 we get Co = -4.

This fits with how we define our components (pg 5). PKA can activate the gene $\log_2=4$ on its own (thus the PKA component is 4), TORC1 can activate the gene $\log_2=4$ on its own (thus the TOR component is 4) but the factors act redundantly so the total expression from both is much less than expected by the mathematical "default" where the pathways act independently--thus there is negative cooperativity or redundancy. It is also important to note that this analysis method can also identify and quantify partial redundancy, in such a case the PKA and TOR components will be larger than the negative cooperative component.

We did not discuss OR gating/redundancy in depth because it does not play a significant role in the TORC1/PKA circuit. As seen in Fig. 2b there are very few genes with a significant negative Co component (some in group II) and in all but a few cases the Co component is small compared to the TOR component. At the Ribi gene module (Group I) there is little redundancy. Instead we see parallel activation by TORC1 and PKA at the early timepoints, and strong AND gate behavior (positive cooperativity) at late time points. To clarify this, we have expanded the section in the legend to Fig. 2a where we go over the forms of regulation and point out that OR gate behavior can be detected by our method (and is present at a low level at the Group II RP genes). This section of the figure legend also points the reader to our previous papers for a detailed discussion of OR gating.

2. The authors derive expression components based on expression changes following drug treatment at various time points following switch to glucose (Fig. 2). The authors should consider including the corresponding expression heatmaps from which these calculations are based. Fig. S3 provides a similar insight. However, it is notable that at 20 min, the independent TOR (and PKA) components contribute substantially (Fig. 2b) whereas "rap vs no drug" elicits relatively little expression change compared to the other treatments (Fig. S3). This is contrary to expectation based on the current model.

The reviewer correctly points out that the impact of TORC1 on gene induction goes down somewhat at 20 min (where PKA activity is very high) compared to at 0 min in Fig. S5 (previously Fig. S3). However, you can also see this in Fig. 2b. The difference between 2b and S5 is partly due to contrast. However, there is also some weak negative cooperativity (redundancy) at this 20 min time-point (light yellow in the Co column). Thus, the TOR component is slightly larger than the change in gene expression observed when rapamycin alone is added to the cell (since the change in rapamycin equals the TOR component plus the negative Co component). We do not bring this up in the text since the impact of the Co component and the small decrease in TORC1 activity is minor, compared to the dominant features of the circuit.

It is important to point out in this context that the expression components contain all of the information in the raw data (see equations legend 2a) but are just presented in a form that make it easier to examine the interaction between the factors. The expression change in rapamycin is the TOR component plus the Co component, the change in 1-NM-PP1 is the PKA component plus the Co component and the change in rapamycin + 1-NM-PP1 are the PKA + TOR + Co components. In other words, there is no missing data when you examine the components instead of the raw data, just a deconvolution of the data (and you can calculate the raw data from the components and vice versa).

3. Both TOR and PKA act on gene expression through phosphorylation of known downstream substrates (Fig. 1a). The authors infer based on drug treatment effects that these pathways contribute to different extents over time. The authors should consider examining changes in downstream phosphorylation markers under these same treatments to assess whether changes in these markers correlate with the expression components inferred by their mathematical model.

This is a good suggestion, but is beyond the scope of this paper. We don't have good assays to follow the phosphorylation of proteins in the RIBI control circuit over time, and would therefore have to use time resolved mass spectrometry to follow the phosphorylation of each TF in the presence and absence of rapamycin and 1-NM-PP1. This would be challenging, especially because the PKA kinases and the TORC1 dependent kinase Sch9 recognize the same (or at least highly overlapping) phosphorylation sites. More importantly, the key result would be to show that PKA phosphorylates the TFs that drive gene expression after 20 min, and then this phosphorylation dies away after 90 and 150 min--but as explained on pg 8 we still don't know what transcription factors are involved in this pulse (its not any of the known factors) and therefore cannot follow their phosphorylation.

MINOR COMMENTS:

4. The authors use NSR1 as a general example of RiBi genes. It may be worthwhile to examine a few other RiBi genes (e.g. RPS or RPL proteins), to confirm on a per gene level that the trend on protein and RNA is replicated, and to examine degree of variation observed. It is worth noting that NSR1 RNA changes might be much higher (approx. 25-fold at peak) than the average RiBi mRNA (approx. 6-fold at peak) (Fig. 1C, 1D), although part of this may be due to the limited dynamic range of microarray measurements.

This a fair point, also made by reviewer 2. We now include time-courses examining mRNA and protein production of two additional Ribi genes—Dhr2 and Mrd1 (Fig. S4). The trends are similar to those found for NSR1.

5. The authors assess RNA levels after 20 min or 30 min drug treatment (Fig. 2, S3), and use this to infer relative contribution of respective pathways. Could the authors provide justification that this is an appropriate length of drug treatment, e.g. perhaps based on RNA degradation rates or similar considerations?

In all experiments, we treated the cells with drugs for 30 min. We were not clear enough about the fact that we added the drug 10 min before we add glucose for the 20min timepoint (to make sure the pathways are off before the large pulse of PKA activity occurs). This is now clarified in the text pg 5. As the reviewer suggests, we choose 30 min in our study to ensure that there is enough time for existing RNA to degrade and reveal the impact that TORC1/PKA have on gene expression. Most mRNAs have a half-life less than 10 min (10) and thus we expect to see at least an 8-fold decrease in gene expression where we completely block transcription (and at least 16-fold for a gene with a 5 min half-life; the average in yeast). If we waited longer, we would see larger changes, but have shown previously (by comparing the impact of TF deletes and CHIP data; (8)), that you also start to see significant secondary (non-specific) effects caused by inhibition of signaling pathways at later timepoints. We have now added text to the end of the section on component analysis to make this clear.

1. Donaton MC, *et al.* (2003) The Gap1 general amino acid permease acts as an amino acid sensor for activation of protein kinase A targets in the yeast *Saccharomyces cerevisiae*. *Mol Microbiol* 50(3):911-929.
2. Huber A, *et al.* (2011) Sch9 regulates ribosome biogenesis via Stb3, Dot6 and Tod6 and the histone deacetylase complex RPD3L. *EMBO J* 30(15):3052-3064.
3. Lippman SI & Broach JR (2009) Protein kinase A and TORC1 activate genes for ribosomal biogenesis by inactivating repressors encoded by Dot6 and its homolog Tod6. *Proc Natl Acad Sci U S A* 106(47):19928-19933.
4. Worley J, Sullivan A, Luo X, Kaplan ME, & Capaldi AP (2015) Genome-Wide Analysis of the TORC1 and Osmotic Stress Signaling Network in *Saccharomyces cerevisiae*. *G3 (Bethesda)* 6(2):463-474.
5. Zaman S, Lippman SI, Schnepfer L, Slonim N, & Broach JR (2009) Glucose regulates transcription in yeast through a network of signaling pathways. *Mol Syst Biol* 5:245.
6. Hughes Hallett JE, Luo X, & Capaldi AP (2014) State transitions in the TORC1 signaling pathway and information processing in *Saccharomyces cerevisiae*. *Genetics* 198(2):773-786.
7. Worley J, Luo X, & Capaldi AP (2013) Inositol pyrophosphates regulate cell growth and the environmental stress response by activating the HDAC Rpd3L. *Cell Rep* 3(5):1476-1482.
8. Capaldi AP (2010) Analysis of gene function using DNA microarrays. *Methods Enzymol* 470:3-17.
9. Capaldi AP, *et al.* (2008) Structure and function of a transcriptional network activated by the MAPK Hog1. *Nat Genet* 40(11):1300-1306.
10. Chan LY, Mugler CF, Heinrich S, Vallotton P, & Weis K (2018) Non-invasive measurement of mRNA decay reveals translation initiation as the major determinant of mRNA stability. *Elife* 7.

Reviewers' comments:

Reviewer #1 (Remarks to the Author):

The revised manuscript by Kunkel et al addressed many of the minor concerns I had but only skirted the major issue, namely does their model apply only to glucose transitions or is it a general model for nutrient transitions. To address my previous critique, they provided data on a nitrogen source upshift and showed that this does induce a transcriptional pulse in *Ribi/rprotein* gene expression. Moreover, they showed that inhibition of TORC1 completely eliminated the spike in expression, whereas inhibition of PKA only partially eliminated it. Accordingly, under the condition of a nitrogen upshift, the TORC1 complex is necessary for the rapid response, which is less dependent on PKA. This is the opposite of the case with glucose upshift. So, I think that they need to revise their conclusion to emphasize that their model applies only to glucose regulation and not to nutrient homeostasis in general.

Reviewer #2 (Remarks to the Author):

The authors have done a good job of responding to reviewers' questions and requests, though they provide relatively little additional experimental data that might lead to mechanistic insights. The study remains a solid but relatively modest contribution to the field. I would make just two points. First, I was not satisfied with the authors' reply to my (admittedly poorly formulated) question regarding *Dot6/Tod6* accumulation during prolonged growth in glucose. The authors' model states that this accumulation promotes stronger *RiBi* down-regulation upon starvation or following TORC1/PKA inhibition. During this period of accumulation, it should be possible to measure increased *Dot6/Tod6* binding if cells are starved at different times during this period (or perhaps even in the absence of starvation or inhibition, since the proteins may be shuttling in and out of the nucleus). A second point is that since the first submission a manuscript describing *Sfp1* action through the RRPE element at *Ribi* and other growth-related genes has appeared (Albert et al. G&D 2019). The authors should consider modifying their depiction of *Sfp1* target genes based upon these new data and commenting on the relevance of this work to their own.

Reviewer #3 (Remarks to the Author):

I have reviewed the authors' responses and the revised manuscript. I found the authors' responses to my previous comments to be satisfactory, and have no further comments.

Reviewer #1 (Remarks to the Author):

The revised manuscript by Kunkel et al addressed many of the minor concerns I had but only skirted the major issue, namely does their model apply only to glucose transitions or is it a general model for nutrient transitions. To address my previous critique, they provided data on a nitrogen source upshift and showed that this does induce a transcriptional pulse in *Ribi*/protein gene expression. Moreover, they showed that inhibition of TORC1 completely eliminated the spike in expression, whereas inhibition of PKA only partially eliminated it. Accordingly, under the condition of a nitrogen upshift, the TORC1 complex is necessary for the rapid response, which is less dependent on PKA. This is the opposite of the case with glucose upshift. So, I think that they need to revise their conclusion to emphasize that their model applies only to glucose regulation and not to nutrient homeostasis in general.

We are glad that the reviewer found the revisions we made acceptable, and agree that we did not fully address the new nitrogen upshift data in the discussion. Therefore, to make it absolutely clear that the model presented in the paper applies to glucose regulation and not nutrient homeostasis in general (or more specifically not to nitrogen signaling) we have added an entirely new paragraph to the discussion. It follows the section where we point out that the TORC1/PKA circuit behaves like a hybrid feedback/feedforward controller. It reads as follows:

*This hybrid feedback/feedforward control model explains the role that the TORC1 and PKA pathways play in regulating gene expression during transitions into and out of glucose, but does not fully explain the response to nitrogen stimuli. Specifically, we find that transferring cells from a poor nitrogen source (proline) to a high-quality nitrogen source (glutamine) triggers a small pulse of *Ribi* gene expression that is completely dependent on TORC1 activity, and only partially dependent on PKA activity (Fig. S6). Thus, it appears that the TORC1 pathway is transiently hyper-activated when cells are first exposed to a high-quality nitrogen source, and consequently that TORC1 has a role outside of simply setting steady-state gene expression levels (at least in response to a nitrogen upshift). Further work is therefore needed to test the hybrid feedback/feedforward control model in a range of conditions and explore its implications.*

We have also carefully examined the rest of the paper to make sure that we don't give the impression that the data presented in this paper (following the glycerol to glucose transition) applies to all nutrient responses—and are confident that it is clear we are only talking about the glucose response.

Reviewer #2 (Remarks to the Author):

The authors have done a good job of responding to reviewers' questions and requests, though they provide relatively little additional experimental data that might lead to mechanistic insights. The study remains a solid but relatively modest contribution to the field. I would make just two points. First, I was not satisfied with the authors' reply to my (admittedly poorly formulated) question regarding Dot6/Tod6 accumulation during prolonged growth in glucose. The authors' model states that this accumulation promotes stronger RiBi down-regulation upon starvation or following TORC1/PKA inhibition. During this period of accumulation, it should be possible to measure increased Dot6/Tod6 binding if cells are starved at different times during this period (or perhaps even in the absence of starvation or inhibition, since the proteins may be shuttling in and out of the nucleus).

As we have discussed at length, we do not believe the Dot6/Tod6 ChIP experiments help us establish any of the key points of this paper since they are redundant with the data we provide showing that Dot6/Tod6 only regulate gene expression when cells are growing in glucose, and are degraded when the cell is starved for glucose. However, we do appreciate the important role that such experiments play in picking apart the mechanistic details of gene regulation and have therefore included a description of our expectations for a Dot6/Tod6 experiment (as requested), and the role that other ChIP and related experiments could have in refining the model we present in the discussion in a paragraph that follows on from the one we added above:

It will also be important to map transcription factor binding to the ribosome and protein synthesis genes over time to determine exactly how the TORC1 and PKA pathways control gene expression. For example, we expect to see an increase in Dot6/Tod6 binding at the Rib genes during the initial response to glucose starvation and then a loss of Dot6/Tod6 binding over time as the factors are degraded. However, it is less clear how Sfp1 and Stb3 binding will change over time, or how these factors will influence gene expression, especially in light of a recent study showing that Sfp1 can bind promoters both directly and through a cofactor⁴².

A second point is that since the first submission a manuscript describing Sfp1 action through the RRPE element at Rib and other growth-related genes has appeared (Albert et al. G&D 2019). The authors should consider modifying their depiction of Sfp1 target

genes based upon these new data and commenting on the relevance of this work to their own.

The new paragraph above also references the new paper on Sfp1 by Albert et al. This paper addresses a long-standing issue in studies looking at Sfp1 —namely that Sfp1 only appears to bind to a subset of the Ribi, RP and other genes that it regulates according to knockout studies. Albert et al finds that while one Sfp1 binding to one set of genes is found by a standard ChIP assay the other genes are identified by creating a Sfp1 micrococcal nuclease hybrid protein and mapping out the cut sites across the genome. While this study is interesting it does not really influence the findings or analysis here since the Sfp1 target genes we examine in Fig. 3 are from the knockout study (not ChIP) and we do not examine the role of Sfp1 in any detail once we show Dot6/Tod6 (and not Sfp1) regulate the genes involved in the TORC1/PKA cooperative response.

Reviewer #3 (Remarks to the Author):

I have reviewed the authors' responses and the revised manuscript. I found the authors' responses to my previous comments to be satisfactory, and have no further comments.

We are glad that Reviewer 3 is satisfied with our previous revisions